# Biology-guided deep learning predicts prognosis and cancer immunotherapy response

Yuming Jiang[1,2,10], Zhicheng Zhang [2,9,10], Wei Wang [3,10], Weicai Huang [1,10], Chuanli Chen[4], Sujuan Xi[5], M. Usman Ahmad [6], Yulan Ren[2], Shengtian Sang[2], Jingjing Xie[7], Jen-Yeu Wang [2], Wenjun Xiong[8], Tuanjie Li[1], Zhen Han[1], Qingyu Yuan[4], Yikai Xu [4], Lei Xing [2], George A. Poultsides[6], Guoxin Li [1,11] ✉ & Ruijiang Li [2,11] ✉

Substantial progress has been made in using deep learning for cancer detection and diagnosis in medical images. Yet, there is limited success on prediction of treatment response and outcomes, which has important implications for personalized treatment strategies. A significant hurdle for clinical translation of current data-driven deep learning models is lack of interpretability, often attributable to a disconnect from the underlying pathobiology. Here, we present a biology-guided deep learning approach that enables simultaneous prediction of the tumor immune and stromal microenvironment status as well as treatment outcomes from medical images. We validate the model for predicting prognosis of gastric cancer and the benefit from adjuvant chemotherapy in a multi-center international study. Further, the model predicts response to immune checkpoint inhibitors and complements clinically approved biomarkers. Importantly, our model identifies a subset of mismatch repair-deficient tumors that are non-responsive to immunotherapy and may inform the selection of patients for combination treatments.

Artificial intelligence, including machine learning and deep learning, is increasingly used to extract information and discover novel patterns from biomedical data[1,2]. These approaches hold an enormous potential for advancing cancer research and clinical care[3]. One of the most successful applications of deep learning is cancer screening for early detection and diagnosis in medical images[4–8]. However, progress has been slower in using deep learning to predict treatment response and

outcomes, which has important implications for personalized treatment strategies[9].

Interpretability is of paramount importance for high-stake clinical applications such as treatment decision-making[2]. Currently the field of deep learning is dominated by a data-driven paradigm, which results in models without intuitive understanding or clear reasoning behind their predictions. This is exacerbated by the fact that prior knowledge

[1]Department of General Surgery, Guangdong Provincial Key Laboratory of Precision Medicine for Gastrointestinal Tumor, Nanfang Hospital, Southern Medical University, Guangzhou, China. [2]Department of Radiation Oncology, Stanford University School of Medicine, Stanford, CA, USA. [3]Department of Gastric Surgery, State Key Laboratory of Oncology in South China, Collaborative Innovation Center for Cancer Medicine, Sun Yat-sen University Cancer Center, Guangzhou, China. [4]Department of Medical Imaging Center, Nanfang Hospital, Southern Medical University, Guangzhou, China. [5]The Reproductive Medical Center, The Seventh Affiliated Hospital of Sun Yat-sen University, Shenzhen, China. [6]Department of Surgery, Stanford University School of Medicine, Stanford, CA, USA. [7]Graduate Group of Epidemiology, University of California Davis, Davis, CA, USA. [8]Department of Gastrointestinal Surgery, Guangdong Provincial Hospital of Chinese Medicine, Guangzhou University of Chinese Medicine, Guangzhou, China. [9]Present address: JancsiTech and Shenzhen Institute of Advanced Technology, Chinese Academy of Sciences, Shenzhen, China. [10]These authors contributed equally: Yuming Jiang, Zhicheng Zhang, Wei Wang, Weicai Huang. [11]These authors jointly supervised this work: Guoxin Li, Ruijiang Li. ✉e-mail: gzliguoxin@163.com; rli2@stanford.edu

about disease biology is ignored during model development. The disconnect from biology leads to models lacking interpretability – a significant hurdle for clinical translation[10]. There is a critical need to incorporate pathobiology into the design of deep learning models to enhance interpretability.

The tumor microenvironment (TME) consists of multiple immune and stromal cell types, which closely interact with each other and contribute to tumor control or progression[11,12]. The prognostic relevance of TME and its impact on treatment response is well established across human cancers[13-16]. In particular, lymphocyte infiltration is associated with favorable outcome and response to immunotherapy;[17,18] while tumor stroma promotes invasion, metastasis, and confers treatment resistance and worse survival[19-22]. Thus, assessment of TME can bring valuable prognostic and predictive information. However, current tissue-based histopathology approach is prone to sampling bias. Imaging, on the other hand, allows the noninvasive characterization of the entire tumor and can be performed repeatedly throughout treatment. Quantitative imaging analysis may reveal the link between radiological phenotypes and the underlying pathobiology[23].

In this work, we propose a biology-guided deep learning framework in which a multi-task model is trained to simultaneously predict TME status and treatment outcomes from radiology images. We perform international validation for prognosis prediction in gastric cancer, one of the most common and lethal malignancies worldwide[24]. We further evaluate the model for predicting survival benefit from chemotherapy and response to immunotherapy.

## Results
### Study design and patient characteristics
The overall study design is shown in Fig. 1. We trained and independently validated a deep learning model that used diagnostic CT images

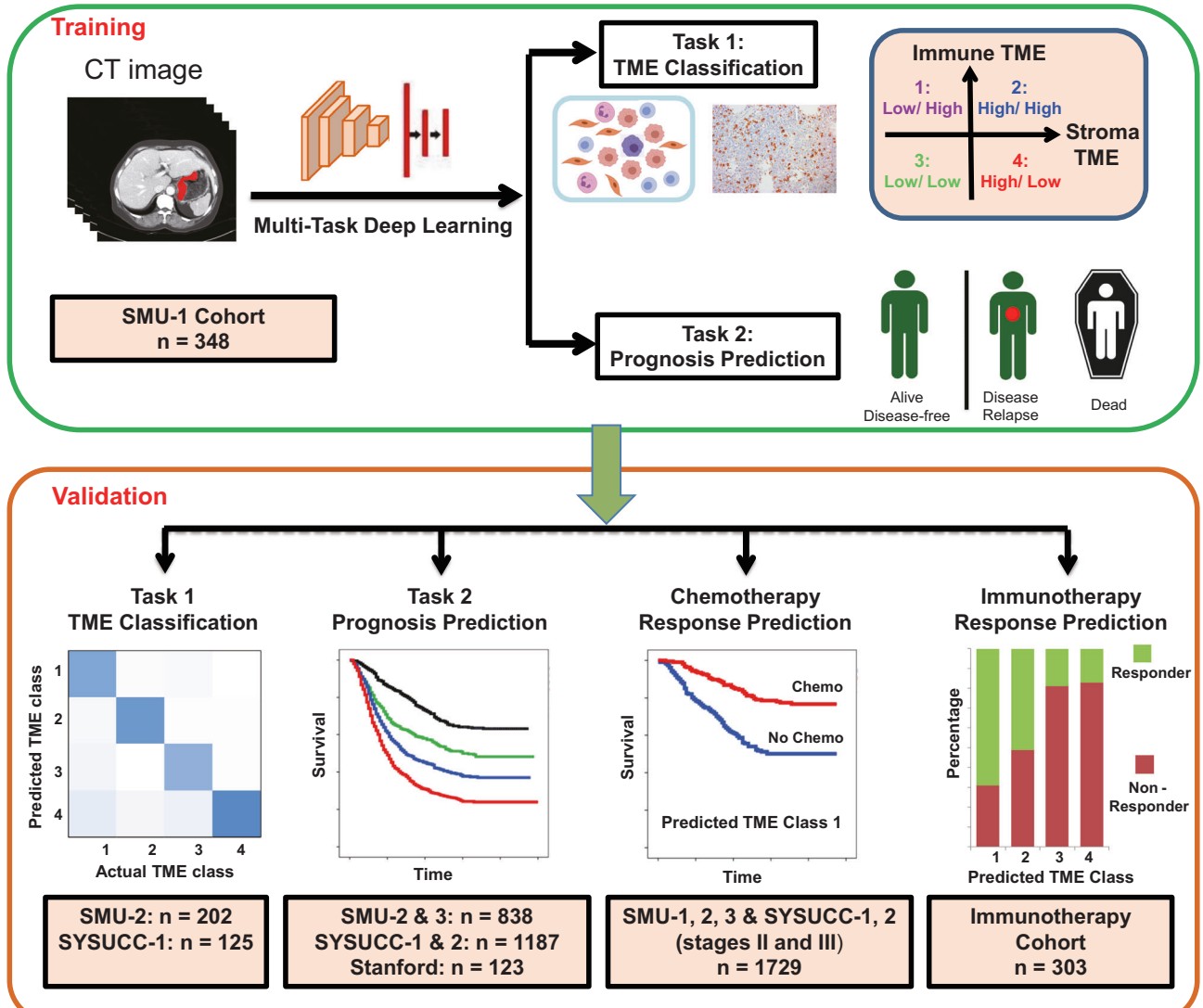

**Fig. 1 | Study design for the development and validation of a deep learning model to predict TME classes and disease-free survival.** Patients in the training (SMU-1) cohort and internal validation (SMU-2, 3) cohorts were recruited from Southern Medical University, Guangzhou, China. Patients in the external validation cohorts were recruited from Sun Yat-sen University Cancer Center (SYSUCC-1, 2), Guangzhou, China and Stanford University Medical Center, Palo Alto, USA. Patients in the immunotherapy cohort were recruited from Southern Medical University, Guangzhou, China and Guangdong Provincial Hospital of Chinese Medicine, Guangzhou, China. Both CT images and IHC-stained slides were available for patients in the SMU-1 training cohort, SMU-2 and SYSUCC-1 validation cohorts, which were used for evaluating the model's accuracy for TME prediction. All patients had preoperative CT scans and outcomes available, which were used for testing the model's prognostic and predictive value. CT: computer tomography; IHC: immunohistochemistry. SMU: Southern Medical University; SYSUCC: Sun Yat-sen University Cancer Center. TME: tumor microenvironment. Chemo: Chemotherapy.

to classify TME and predict prognosis of patients with gastric cancer. The rationale for combining TME and outcome prediction in a single model is that they are closely related and inter-connected tasks given the established mechanistic link between the two. We hypothesize that this approach could lead to improved generalizability with the added benefit of enhanced interpretability. We tested the model for its ability to predict benefit from adjuvant chemotherapy in non-metastatic disease as well as to predict immunotherapy response in advanced gastric cancer.

We recruited patients in four academic medical centers from China and United States (Supplementary Fig 1). A total of 2799 patients met inclusion criteria and were divided in to in 7 cohorts. Among these, 2496 patients in 6 cohorts were treated with surgery with or without adjuvant chemotherapy, and the clinicopathological characteristics are listed in Supplementary Data 1. The majority of these patients ($n = 1806$, 72.36%) had stage II or III disease, and 928 (51.38%) patients

received adjuvant chemotherapy. In the 7th cohort, we included 303 patients who received anti-PD-1 immunotherapy (Supplementary Data 2). All patients had stage IV gastric cancer, and most (94%) tumors were mismatch repair deficient (dMMR) or MSI-H.

## Development of the biology-guided deep learning (BgDL) model

We trained a multi-task deep convolutional neural network model based on CT image to simultaneously classify TME status and predict prognosis (Fig. 2A). Here, four TME classes are defined based on immunohistochemistry (IHC) evaluation of established immune and stromal markers (see Methods and Supplementary Table 1); and primary clinical endpoint is disease-free survival. The implementation of the deep learning model is available at: https://github.com/zzc623/ClassGastric.

Figure. 2B shows the CT images and corresponding feature maps along with the predicted TME classes and survival scores for four

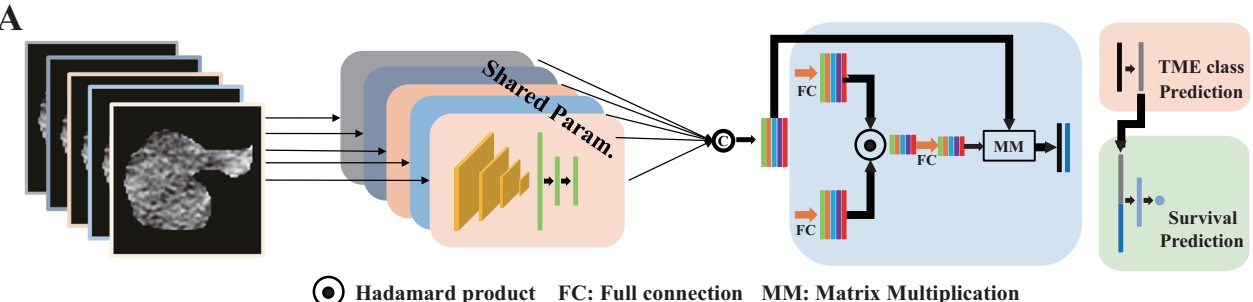

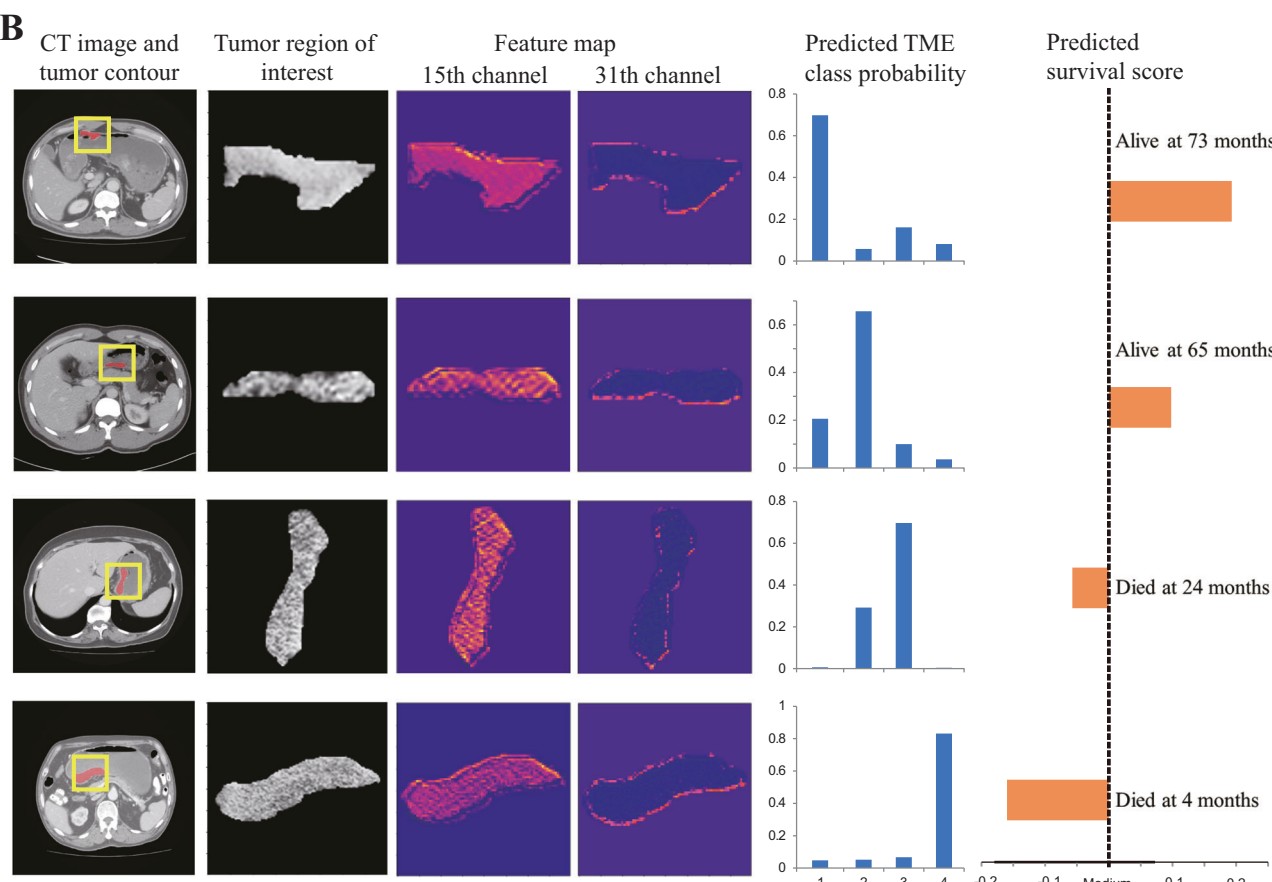

**Fig. 2 | Proposed deep learning model and visualization, prediction for representative cases. A** Architecture of the proposed multi-task deep convolutional neural network to simultaneously classify TME and predict prognosis from CT image; (**B**) CT images and corresponding feature maps along with the predicted

TME classes and survival scores for four representative cases, where each row corresponds to a patient with TME classes 1–4 defined by IHC. TME classes were correctly predicted for all four cases; predicted survival scores were also consistent with the actual patient outcome. TME tumor microenvironment.

representative cases. Visually the feature maps appear to be related to tumor heterogeneity and imaging characteristics of the invasive margin. To understand what type of information the feature maps represent, we computed radiomics features based on the feature maps in the 15th channel using the PyRadiomics package[25]. We observed significant differences in the texture feature values among the four TME classes. Features that measure heterogeneity generally show increasing patterns from TME class 1 through 4, while those measuring homogeneity show decreasing patterns (Supplementary Fig 2). This indicates that the deep learning model may capture important information related to intratumor heterogeneity. Our result is consistent with previous findings that tumor heterogeneity is associated with a worse prognosis[26].

The relationships between the deep learning model predicted TME classes, survival score and clinicopathological variables are summarized in Supplementary Data 3–10. The model predictions were statistically associated with certain risk factors, most notably pT stage. However, there was no clear-cut relation between the two, and substantial overlap exists in predicted survival scores across stage subgroups. The magnitude of correlation between survival score and tumor size was low (Pearson correlation coefficients 0.14–0.17).

### BgDL model accurately classifies the tumor immune and stroma microenvironment

The proposed deep learning model achieved a high accuracy for classifying the four TME classes in the training cohort (Fig. 3A). This model showed similarly high levels of discrimination, with AUCs of 0.94–0.96 and 0.94–0.97 in the internal and external validation cohorts, respectively (Fig. 3A). The specificity and negative predictive value were above 90%, while the sensitivity and positive predictive value varied from 70–93% to 80–88% in the validation cohorts (Supplementary Table 2). Consistently, the confusion matrix showed that the model predictions agreed well with the actual TME classes defined by IHC (Fig. 3B). The overall accuracy for the four TME classes was around 0.810 (95% CI: 0.769–0.851), 0.832 (0.780–0.884) and 0.840 (0.776–0.904) in the training and two validation cohorts.

Given the prognostic relevance of TME, we first confirmed that the four TME classes defined according to IHC were associated with distinct prognoses including DFS and OS (Supplementary Fig 3). We then assessed the relation between the model predicted TME classes and prognosis. Indeed, the same pattern was observed across the training and all validation cohorts (Supplementary Fig 4).

### BgDL model predicts prognosis independently of clinicopathologic factors

Based on the BgDL model-predicted survival scores, we divided patients into two risk groups by using the medium value (−0.50) in the training cohort and applied the same cutoff point to all validation cohorts. We observed a significant difference in both DFS and OS between patients with low vs. high score in the training cohort ($P < 0.0001$; Fig. 4A). The 5-year DFS and OS rates for high-risk patients were 19.83% and 20.66%, and for low-risk patients were 43.61% and 54.63%, respectively (Fig. 4A). The same pattern between model-predicted survival scores and prognosis was observed consistently across all 5 independent validation cohorts (all $P < 0.0001$; Fig. 4B–F).

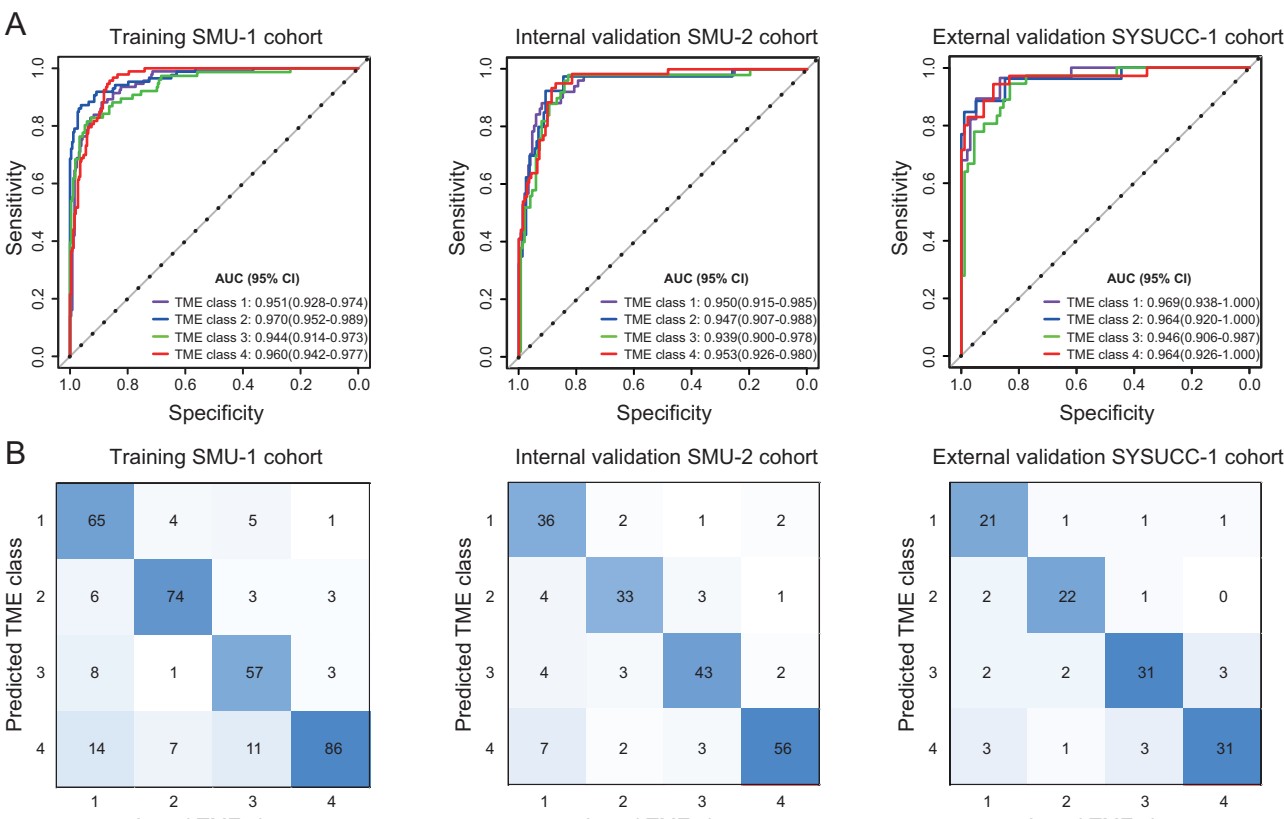

**Fig. 3 | Accuracy of the deep learning model to assess TME classes. A** Receiver operator characteristic (ROC) curves and (**B**) confusion matrices in the training SMU-1 cohort, internal validation SMU-2 cohort, and external validation SYSUCC-1 cohort. The ROC curves show the one-vs-others comparison. The confusion matrices show the pair-wise comparison; diagonal: number of cases correctly classified; off-diagonal: number of cases incorrectly classified. TME tumor microenvironment, SMU Southern Medical University, SYSUCC Sun Yat-sen University Cancer Center, AUC area under the curves. Source data are provided as a Source Data file.

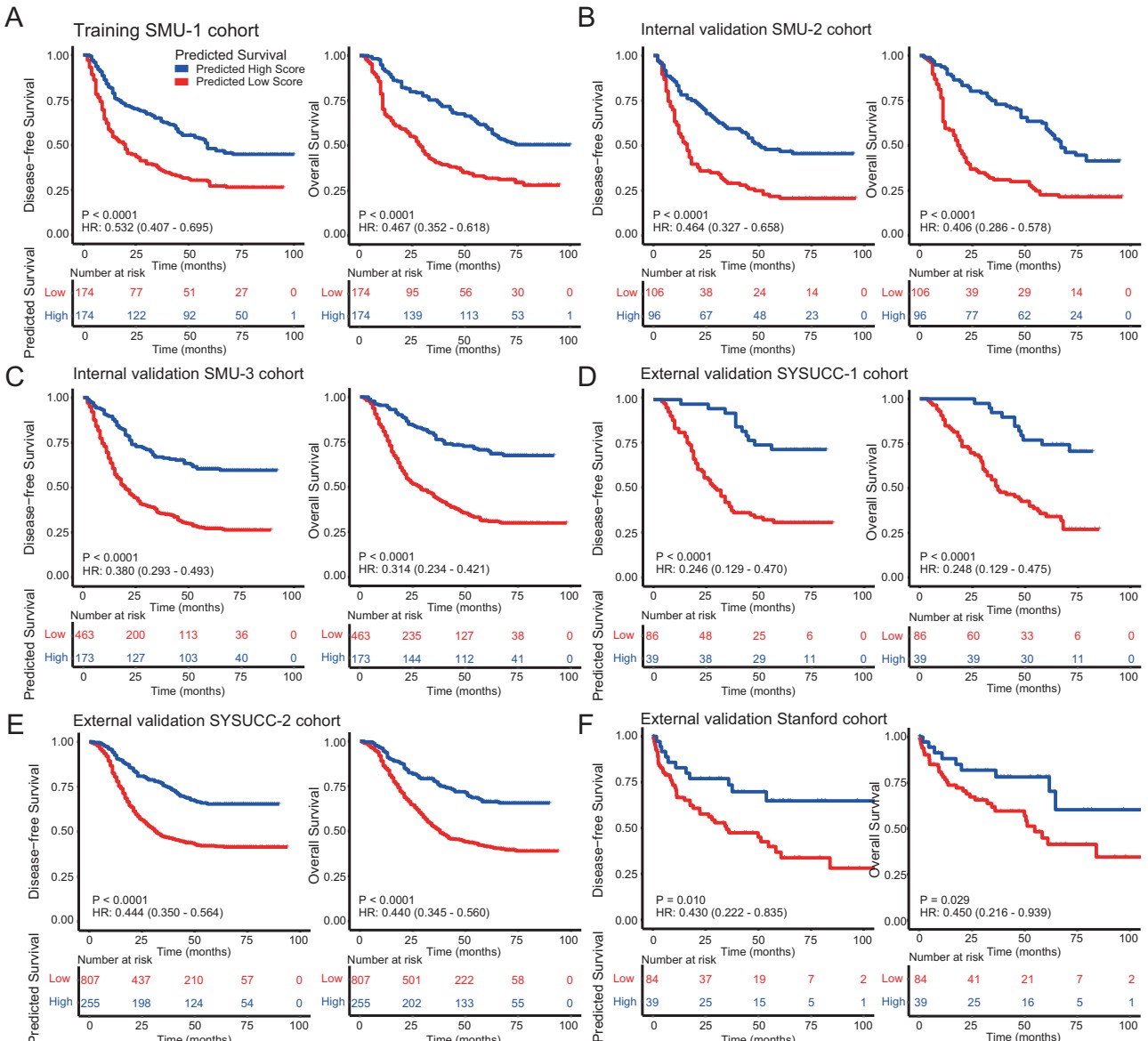

**Fig. 4 | Kaplan-Meier analyses of disease-free survival (DFS) and overall survival (OS) according to the model-predicted survival score in patients with gastric cancer. A** Training cohort SMU-1 ($n = 348$), (**B**) SMU-2 validation cohort ($n = 202$), (**C**) SMU-3 validation cohort ($n = 636$), (**D**) SYSUCC-1 validation cohort ($n = 125$), (**E**) SYSUCC-2 validation cohort ($n = 1063$), (**F**) Stanford validation cohort ($n = 123$). Comparisons of the survival curves were performed with a two-sided log-rank test. SMU Southern Medical University, SYSUCC Sun Yat-sen University Cancer Center, HR Hazard ratio. Source data are provided as a Source Data file.

We compared the discrimination performance of the proposed deep learning model with traditional clinicopathologic risk factors in terms of AUC for prognosis prediction. Our deep learning survival score (DLS) showed superior discriminability to almost all risk factors including pT, pN, pM stage, tumor size, differentiation, Lauren histology, serum CEA and CA199 levels (Fig. 5A). The only exception was overall pathologic stage, which had a similar performance with the deep learning model for prediction of 5-year DFS (AUC: 0.72 vs. 0.71).

We then performed multivariate Cox regression analysis adjusting for clinicopathologic factors. The DLS model remained an independent prognostic factor for both DFS and OS in the training cohort and validation cohorts (all $P < 0.0001$; Supplementary Data 11; Supplementary Tables 3–8). We also assessed the relative contribution of individual variables for prognosis prediction. Among the clinicopathologic risk factors, pT, pM, and pN stage were the most important variables. However, when DLS was added to the model, it became the most important parameter for prognosis prediction (Fig. 5B). For patients with stage II or stage III disease, DLS was the dominant variable for prognosis prediction, accounting for 76.5% and 49.6% in relative contribution (Fig. 5B). Similar results were also observed for OS (Supplementary Fig 5).

We further assessed the prognostic value of the deep learning model in each subgroup of patients with the same clinicopathologic risk factors. DLS significantly stratified patients for DFS in all stage subgroups (Fig. 5C). Strikingly, stage II patients with high DLS had an even better overall survival compared with stage I patients with low DLS, HR = 0.921 (0.856–0.991), $P = 0.028$ (Supplementary Fig 6); a similar trend exists for DFS, HR = 0.943 (0.880–1.101), $P = 0.101$. DLS also stratified patients for DFS and OS within each subgroup of T stage, N stage, tumor size, grade, histologic subtype, etc (Supplementary Figs. 7–8). Although the training cohort consists of only Asian patients, DLS significantly stratified for DFS and OS among patients with non-Asian ethnicity in the Stanford cohort (Supplementary Fig 9).

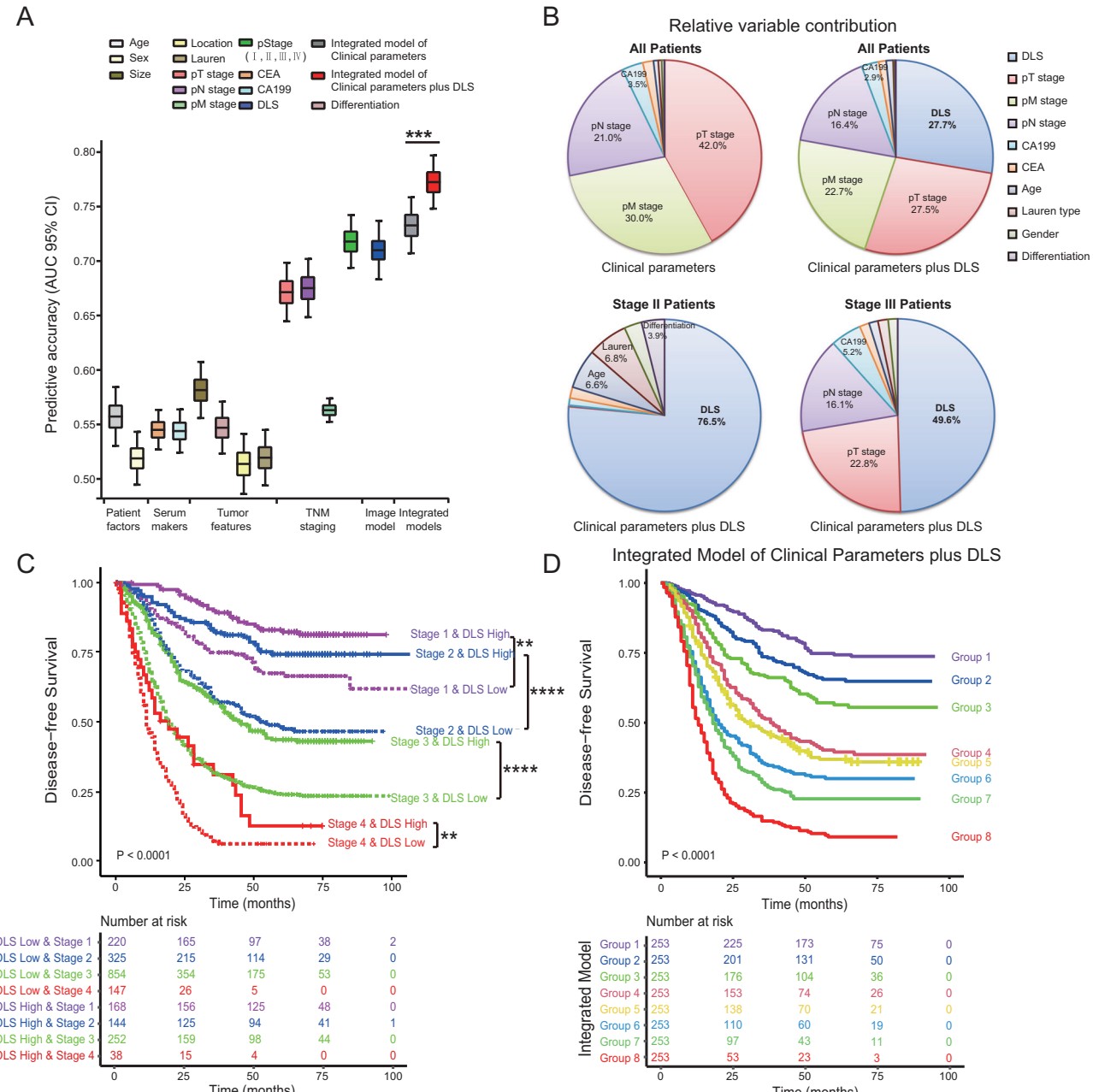

**Fig. 5 | Prognosis prediction using the deep learning model and clinicopathologic risk factors. A** Accuracy of prediction for disease-free survival (DFS) using clinicopathologic variable and the deep learning model ($n = 2025$). The center lines within the boxes represent the mean AUC value, the bounds of boxes represent the interquartile range (IQR) and the whiskers represent the 95% confidence intervals. **B** Relative variable contribution to prediction of DFS using the $\chi^2$ proportion test for clinicopathologic variables only in all patients ($n = 2025$); for clinicopathologic variables and DLS in all patients as well as patients with stage II and III disease. **C** Kaplan-Meier analysis of DFS according to the deep learning model-predicted survival score within each stage in the validation cohorts ($n = 2148$). **D** Kaplan-Meier analysis of DFS according to the nomogram combining deep learning model and clinicopathologic risk factors in the validation cohorts ($n = 2025$). For statistical comparisons among different groups, a two-tailed $t$ test (unpaired) was used. Comparisons of the survival curves were performed with a two-sided log-rank test. DLS deep learning survival score. **$P < 0.001$, ***$P < 0.0001$, ****$P < 0.00001$. Source data are provided as a Source Data file.

## BgDL model improves risk stratification when integrated with clinicopathologic factors

Since the imaging-based deep learning model demonstrated complementary prognostic value to clinicopathologic risk factors, we combined these multi-modal data to build an integrated model for individualized assessment of prognosis in the training cohort (Supplementary Fig. 10). Compared with a model consisting of only clinicopathologic factors, the integrated model significantly increased the AUC (0.77 vs. 0.73, $P < 0.001$) for prognosis prediction in the validation cohorts (Fig. 5A). In addition, we calculated the C-index for the deep

learning model, clinicopathologic variables, and integrated models. The results show similar patterns when comparing different variables and models (Supplementary Fig. 11). Remarkably, the integrated model can stratify patients into 8 different risk groups, each with a distinct prognosis (Fig. 5D and Supplementary Fig. 12). The calibration curves at 1, 3, or 5 years showed good agreement between the model estimations and the actual observations for probabilities of DFS (Supplementary Fig. 13). Consistently, the corresponding prediction error curves showed a lower prediction error for the integrated model compared with stage, clinicopathologic model, and the imaging model

                                                                    

(Supplementary Fig. 14). Similar results were obtained for integrated Brier score (IBS) with lower values indicating better performance (Supplementary Table 9).

We then quantified the relative improvement in survival prediction accuracy for the integrated model vs. clinicopathologic model. This analysis led to a net reclassification improvement (NRI) of 0.156 (0.005–0.286; $P = 0.030$) and 0.156 (0.005–0.286; $P = 0.010$) for DFS and OS in the internal validation cohort; NRI of 0.187 (0.064–0.272; $P < 0.001$) and 0.187 (0.068–0.277; $P < 0.001$) for DFS and OS in the external validation cohort, respectively (Supplementary Table 10). Similar results were also found in comparing the integrated model with the imaging model alone (Supplementary Table 10).

## BgDL model outperforms traditional deep learning approach

We compared the proposed multi-task deep learning model for simultaneous TME classification and survival prediction with the traditional approach, in which two deep learning models (with the same network architecture) were trained separately for the two different tasks. In both validation cohorts, the proposed biology-guided deep learning significantly improved the performance of prognosis prediction with AUCs 0.716–0.754 vs. 0.670–0.672 compared with the traditional biology-naïve approach (Supplementary Fig. 15B). Importantly, multi-task deep learning also achieved superior performance in survival prediction than models solely based on the estimated TME with AUCs 0.658–0.677 (Supplementary Fig. 16), suggesting that the model captures prognostically relevant information beyond TME. In addition, multi-task learning also improved the performance of TME classification with overall accuracy 0.832–0.840 vs. 0.757–0.760 (Supplementary Fig. 15A).

## BgDL model predicts survival benefit from adjuvant chemotherapy

After validating the deep learning model for TME classification and prognosis prediction, we further evaluated its relation to survival outcomes in patients who either received or did not receive adjuvant chemotherapy. This analysis was performed in patients with stage II and III disease, for which an individualized decision of chemotherapy would be most beneficial. To mitigate potential selection bias in our retrospectively data, we performed 1:1 propensity score matching within each TME class between patients who were treated with and without chemotherapy. After matching, most clinicopathological characteristics were similar between the two groups (Supplementary Data 12–13).

For patients in the predicted TME class 1 group, chemotherapy was associated with significantly improved DFS, HR = 0.258 (0.172–0.388), $P < 0.001$ (Fig. 6A). For patients in the predicted TME class 2 group, chemotherapy was also associated with improved DFS although with a smaller effect (HR = 0.691 (0.552–0.914), $P = 0.009$, Fig. 6A). By contrast, chemotherapy was associated with worse DFS in the predicted TME class 4 group (HR = 1.669 (1.363–2.043), $P < 0.001$). Chemotherapy did not appear to have any effect on DFS in the predicted TME class 3 group (HR = 0.831 (0.656–1.053), $P = 0.120$).

Considering the modest effect size of chemotherapy in predicted TME classes 2 and 3, we incorporated the information about prognosis to further assess their relation to chemotherapy benefit. We found that for patients with low DLS score, chemotherapy was associated with significant improvement in DFS in both TME classes 2 and 3 groups (Fig. 7A, B). On the other hand, for patients with high DLS score, chemotherapy did not have any impact on DFS in either group (Fig. 7A, B). We performed a formal statistical interaction test between the model predictions and chemotherapy, which confirmed a significant interaction ($P < 0.05$) regarding the impact on DFS and OS in TME classes 2 and 3 group (Fig. 7A, B and Supplementary Fig 17). Further, we performed multivariate logistic regression analysis and confirm that the imaging based TME class is indeed an independent factor for predicting the benefit of adjuvant chemotherapy in gastric cancer (Supplementary Table 11).

The effects of chemotherapy in different subgroups of patients are summarized in Fig. 6B and Fig. 7C. This analysis shows a clear survival benefit from chemotherapy in patients with predicted TME class 1 as well as TME classes 2 and 3 with low DLS score. Our result also indicates a lack of benefit from chemotherapy in patients with predicted TME class 4. We repeated the above analyses using all the patients without propensity score matching and obtained similar results (Supplementary Figs. 18–19). These data suggest that the deep learning model may be predictive of the benefit of adjuvant chemotherapy in stage II and III disease.

## BgDL model predicts response to anti-PD-1 immunotherapy

We finally investigated relations between the deep learning model (in particular, predicted TME classes) and response to anti-PD-1 immunotherapy in advanced gastric cancer. In a cohort of 303 0patients, the overall objective response rates were 34.7%. For patients in the predicted TME classes 1 and 2 groups, the objective response rates were substantially higher (69.0% and 53.3%, respectively) compared with those in the predicted TME classes 3 and 4 groups (18.4% and 17.2%, respectively) (Fig. 8A). For all patients, the median progression-free survival (PFS) was 8.5 months. Kaplan–Meier analysis showed that the predicted TME classes were significantly associated with PFS ($P < 0.001$; Fig. 8B). The median PFS was 25.0, 18.0, 7.0, 5.0 months in patients with predicted TME classes 1, 2, 3, 4, respectively.

Combined Positive Score (CPS) of PD-L1 expression, which is an approved biomarker of immunotherapy response, was also significantly associated objective response (Supplementary Fig. 20). However, the predictive accuracy for CPS was quite modest (AUC: 0.646 (95% CI 0.580–0.713); Fig. 8C, D). In comparison, the predicted TME classes showed a higher accuracy for predicting objective response (AUC: 0.753 (0.692–0.814); Fig. 8C). Consistently, TME classes had a stronger effect on objective response than CPS in multivariate regression analysis (Fig. 8E). Further, TME classes can distinguish patients with differential response within the CPS moderate and high subgroups (Supplementary Fig. 21), suggesting a complementary relation between the two.

Since these parameters reflect different aspects of tumor biology, we developed an integrative decision tree model (Fig. 8F) that combined CPS and TME classes, which significantly improved the response prediction accuracy (AUC: 0.806 (0.753–0.859); Fig. 8D) compared with CPS. Of note, for patients with high CPS, as much as 80% of patients with TME class 1 had an objective response. In comparison, only one third of patients with TME class 3 and 4 responded to immunotherapy, which means that two thirds of patients did not respond despite having high CPS. On the other hand, patients with low CPS generally had a low response rate (as low as 10% in TME class 3 and 4). However, over 40% of patients with TME class 1 still responded to immunotherapy despite having low CPS (Fig. 8F).

Finally, it is important to evaluate the predictive values of the TME classes in different therapeutic settings. We therefore analyzed separately 68 patients treated with single-agent immunotherapy and 235 patients treated with combination chemoimmunotherapy. Our analyses show that the TME classes achieved similarly good predictive value in either therapeutic setting, and integration with CPS further improved the accuracy for predicting treatment response (Supplementary Fig. 22, 23).

## Discussion

In this work, we developed a biology-guided deep learning approach that allows the simultaneous prediction of the immune and stromal tumor microenvironment (TME) status as well as prognosis from radiology images. We extensively validated the model for prognosis prediction in an international multi-center cohort of 2799 patients with

                                                                    

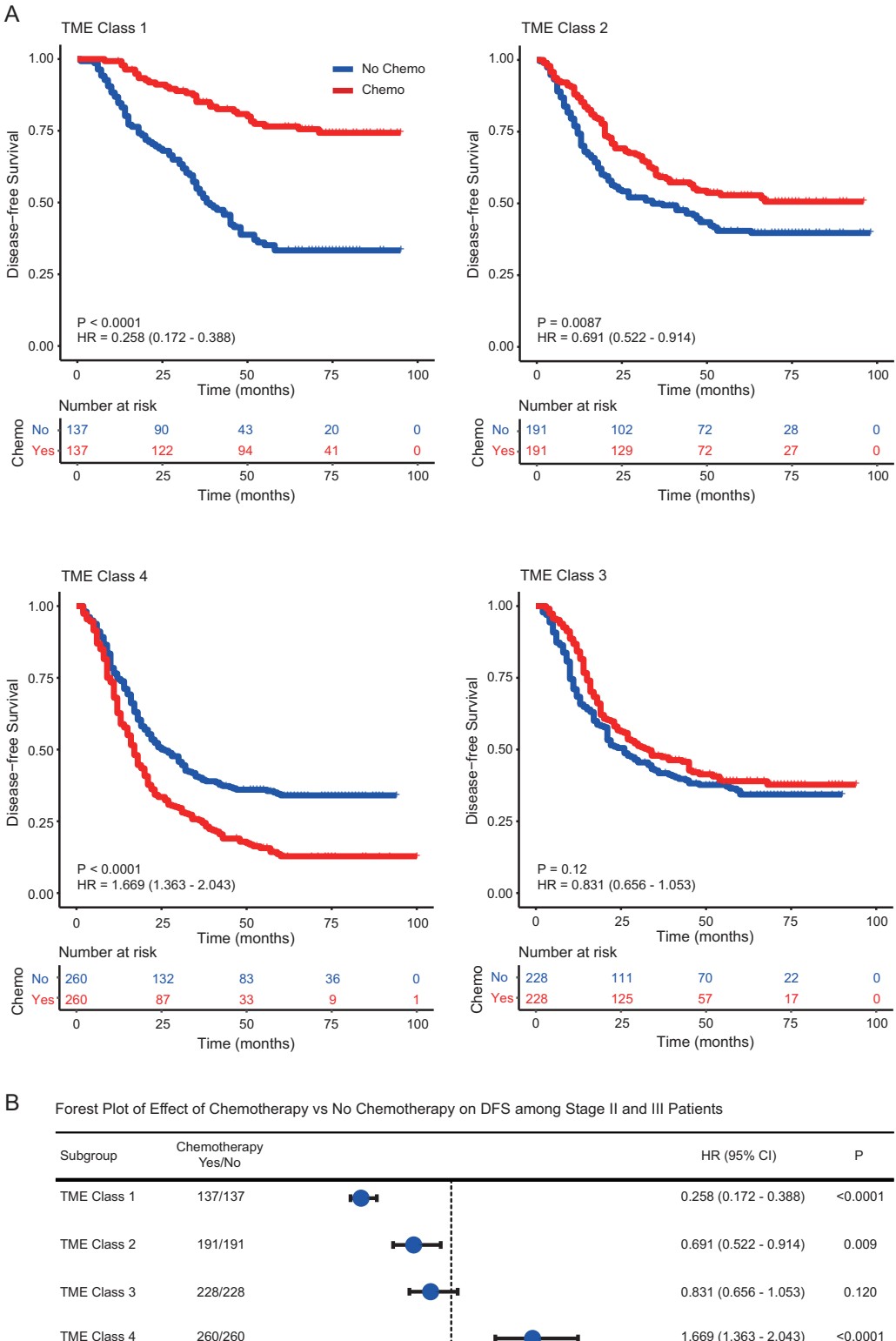

**Fig. 6 | Relationship between the TME class groups and benefit from adjuvant chemotherapy in matched patients with stage II and III gastric cancer.** Kaplan-Meier curves of disease-free survival (DFS) for patients stratified by the receipt of chemotherapy. **A** TME class 1 (*n* = 274), TME class 2 (*n* = 382), TME class 3 (*n* = 456), TME class 4 (*n* = 520). **B** Forest plot for the effect of chemotherapy vs. no chemotherapy on DFS among stage II and III patients. Comparisons of the above survival curves were performed with a two-sided log-rank test. *P* values reported in (**B**) are two-tailed from Cox proportional hazard regression analyses. Blue dot represents the HR value. Error bars represent the 95% confidence intervals. TME tumor microenvironment, DLS deep learning survival score, TME tumor microenvironment, Chemo Chemotherapy, HR, Hazard ratio. Source data are provided as a Source Data file.

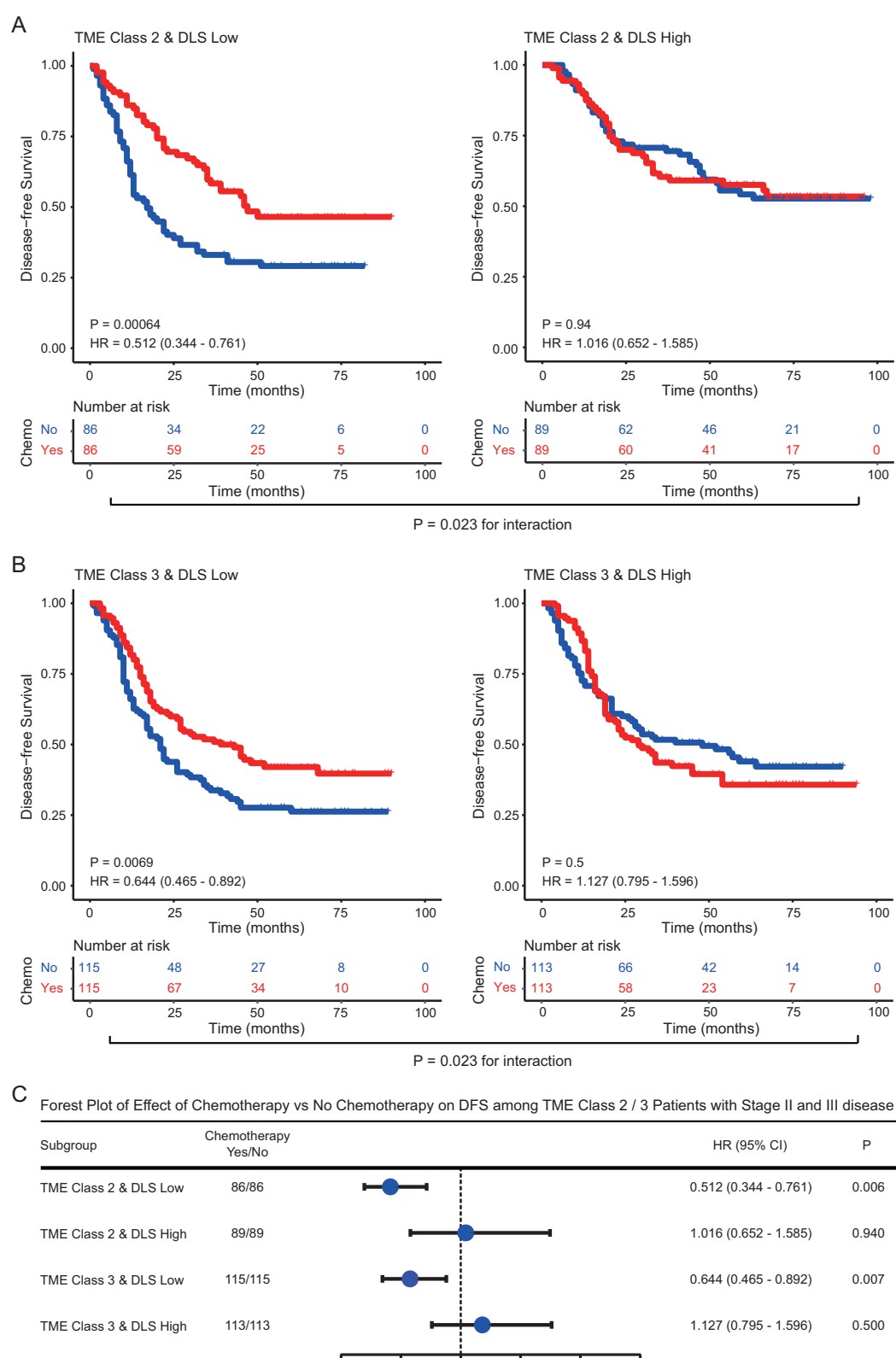

**Fig. 7 | Relationship between the deep learning model and benefit from adjuvant chemotherapy in matched patients with stage II and III gastric cancer.** Kaplan-Meier curves of disease-free survival (DFS) for patients stratified by the receipt of chemotherapy. **A** TME class 2 & DLS Low (*n* = 172), TME class 2 & DLS High (*n* = 178). **B** TME class 3 & DLS Low (*n* = 230), TME class 3 & DLS High (*n* = 226). **C** Forest plot for the effect of chemotherapy vs. no chemotherapy on DFS among TME Class 2/3 patients with stage II and III disease. Comparisons of the above survival curves were performed with a two-sided log-rank test. *P* values reported in (**C**) are two-tailed from Cox proportional hazard regression analyses. Blue dot represents the HR value. Error bars represent the 95% confidence intervals. DLS deep learning survival score, TME tumor microenvironment, Chemo Chemotherapy, HR Hazard ratio. Source data are provided as a Source Data file.

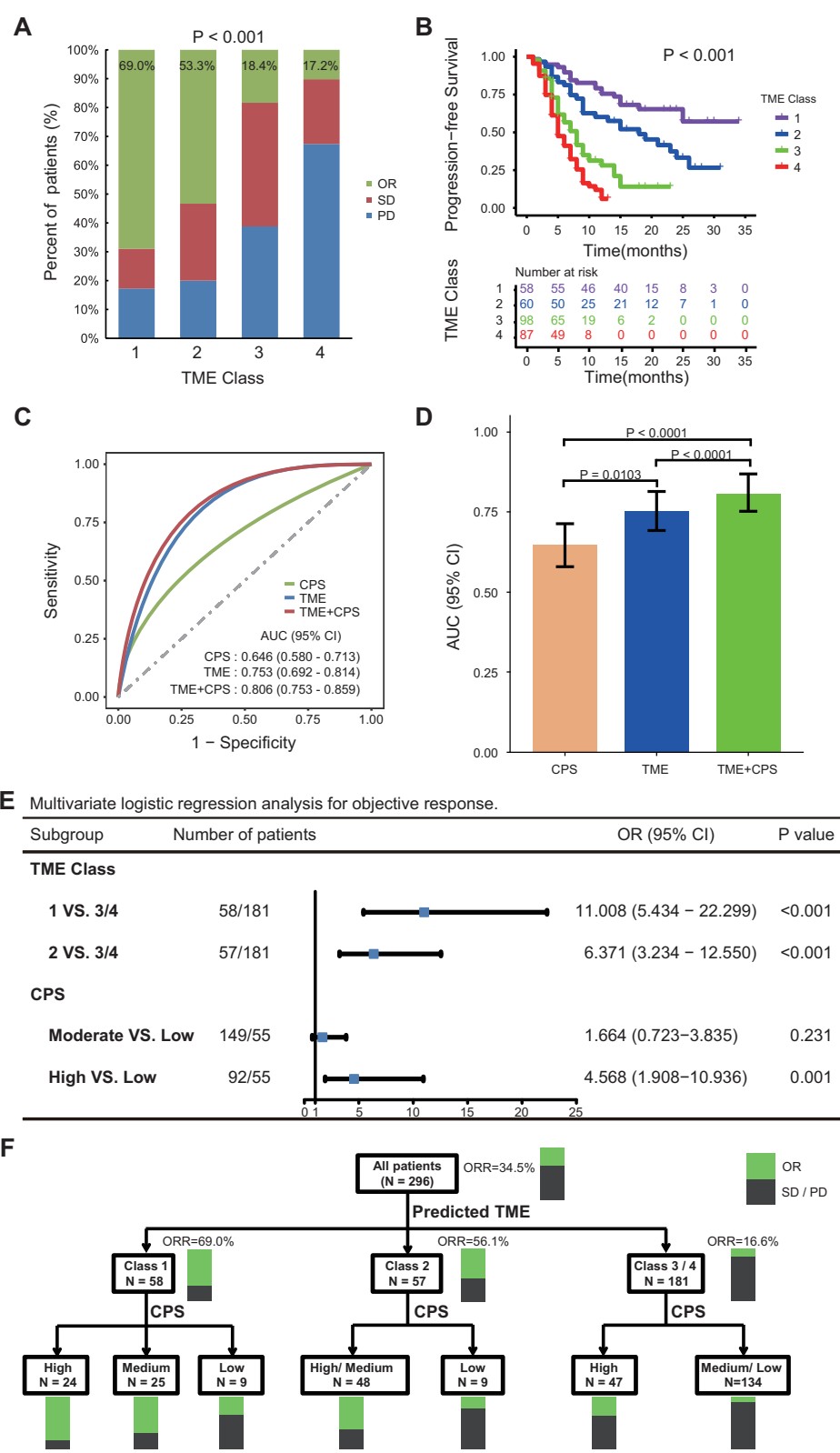

gastric cancer. Importantly, the model could identify a subset of patients who would derive survival benefit from adjuvant chemotherapy. Further, we showed the model could predict response to immunotherapy and combination with an existing biomarker led to a significant improvement in prediction accuracy.

Despite the success and enormous potential of deep learning, interpretability remains one of the most significant challenges for clinical translation. This is particularly true for high-stake applications such as treatment decision-making. Unfortunately, current deep learning models are purely data-driven and do not take into account prior knowledge about the disease pathobiology. The lack of interpretability diminishes trust and may contribute to limited reproducibility and generalizability. To address this issue, we propose a approach to incorporate biological knowledge into the model ab initio,

**Fig. 8 | Performance of the deep learning model in predicting response and outcomes in patients treated with anti-PD-1 immunotherapy. A** Response rates in patients of four TME classes predicted by the deep learning model; (**B**), Progression-free survival in patients of four predicted TME classes; (**C**), Receiver operator characteristic (ROC) curves of the predicted TME classes, CPS and composite models combining TME classes and CPS for predicting immunotherapy response (*n* = 296); (**D**), AUC values of the predicted TME classes, CPS and composite models combining TME classes and CPS for predicting immunotherapy response (*n* = 296); (**E**), Forest plot for the multivariate logistic regression analysis for objective response; (**F**), Decision tree combining the predicted TME classes and CPS. Comparisons of the survival curves were performed with a two-sided log-rank test. Comparisons of the bar plot were performed with a two-sided *t* (unpaired) test. *P* values reported in (**E**) are two-tailed from logistic regression analyses. Blue dot represents the HR value. Error bars in (**D**) and (**E**) represent the 95% confidence intervals. TME tumor microenvironment, AUC area under the receiver operator characteristic curve, CPS combined positive score of PDL1 expression, OR objective response (complete and partial response), SD stable disease, PD progressive disease. Source data are provided as a Source Data file.

via multi-task learning for the simultaneous prediction of prognosis and tumor microenvironment. This contrasts with previous models that attempt to perform '*post hoc*' explanation via saliency or attention maps[27]. We show that incorporation of biological domain knowledge as integral components of deep learning not only improved generalizability compared with the traditional approach but also enhanced interpretability of the model.

Our work builds on extensive evidence for the well-established role of TME in disease progression and impact on treatment response and resistance[13–16]. The radiological approach provides a noninvasive means to the evaluation of immune and stroma TME, which may complement histological evaluation based on tissue specimen. While histological assessment remains the gold standard, this approach is limited by an insufficient amount of tissue practically available (especially in small biopsies) and is prone to sampling bias due to intratumor spatial heterogeneity[28]. On the other hand, radiological imaging allows noninvasive, unbiased evaluation of the entire tumor, and can be acquired repeatedly throughout the treatment. Since TME is dynamic and may evolve with disease progression or treatment, this approach also opens the door for longitudinal monitoring of TME.

Several groups including ours have developed radiomic signatures of immune biomarkers that were subsequently correlated with treatment response and outcomes[29–33]. Here, we included both lymphoid/myeloid immune cells and stromal components to capture the complexity and heterogeneity of TME across patients. Instead of using a sequential design[31,32], we propose a multi-task deep learning strategy for concurrent prediction of treatment outcome and TME status. By incorporating domain knowledge as integral components of the network and learning shared representations, this approach has the dual advantages of using biology to guide outcome prediction and enabling more data-efficient training. Since the model is trained to simultaneously predict outcome, the model captures prognostically relevant information beyond TME. Indeed, as shown in our results, this led to improved performance over TME-based prognosis prediction.

The imaging-based deep learning model outperformed individual clinicopathologic factors and achieved a prognostication accuracy that is on par with the overall disease stage. Given its complementary value, the model may be used to refine current staging system and improve risk stratification of gastric cancer. One strength of our work is that the prognostic model was validated in broad populations from different geographic locations including East Asia and North America and is generalizable across racial groups. Given the wide availability and relatively low cost of CT, the imaging-based model may have a broad clinical impact, especially on underserved populations.

Our results indicate a predictive effect of the deep learning model for the benefit of adjuvant chemotherapy in gastric cancer. Patients with the predicted high immune/low stroma TME would clearly derive survival benefit from chemotherapy. On the other hand, patients with predicted low immune/high stroma TME would not benefit and may even be harmed by chemotherapy. Interestingly, for patients with alternative TME classes, the effect of chemotherapy depends on their predicted prognosis: only patients with predicted worse prognosis seem to benefit while others do not benefit from chemotherapy. For these patients, more intensive therapies will be needed to overcome treatment resistance and improve their outcomes[34,35].

We showed that the deep learning model of TME classes could predict response to anti-PD-1 immunotherapy in advanced gastric cancer. Specifically, tumors with high stroma TME had poor response to immunotherapy regardless of the immune status. On the other hand, tumors with high immune TME showed good response in the context of low stroma TME. This is generally consistent with previous findings based on molecular approaches to TME evaluation[36]. The deep learning model of TME had a stronger effect than PD-L1 expression, an approved biomarker of immunotherapy response. Importantly, a simple and interpretable model that combines TME classes with PD-L1 expression achieved a significantly higher accuracy for response prediction. Of note, our analysis revealed that a specific group of patients with dMMR/MSI-H tumors remain unresponsive to anti-PD-1 immunotherapy. This finding has clinical implications since these patients are recommended to receive immunotherapy under current treatment guidelines, and novel combination therapies will be necessary to improve response rates.

Our study is mainly limited by the retrospective nature and potential selection bias. To mitigate this problem, we performed rigorous validation in a large international dataset and adjusted for clinicopathologic factors in our analysis. At present, manual delineation of the tumor is required for the best model performance. This is mainly because with current deep learning it is challenging to reliably segment gastric cancer due to the inherent nature of these tumors, such as irregular shapes and indistinct boundary present in the CT images. In future, development of more advanced methods for automated or semi- automated tumor segmentation can help facilitate practical implementation of this approach.

Although we focused on specific aspects of the tumor microenvironment in this work, the proposed framework is general in that any relevant information about cancer biology can be incorporated to design knowledge-guided deep learning models for prediction of treatment response and outcomes. As an alternative approach to deep learning, radiomics that are based on user-defined computational image features may offer some degree of interpretability[26,37–39]. In future work, it may be beneficial to combine the two complementary approaches to further enhance model performance and interpretability[40].

In conclusion, we present a biology-guided deep learning model that improves the prediction of treatment response and outcomes in gastric cancer using radiology images. The proposed concept is broadly applicable to other tumor types and may afford a noninvasive approach for evaluation of the tumor microenvironment to inform personalized cancer therapy.

## Methods
### Patients and data collection
This study was approved by the Institutional Review Board at four academic medical centers, including Nanfang Hospital of Southern

Medical University, Sun Yat-sen University Cancer Center, Guangdong Provincial Hospital of Chinese Medicine, and Stanford University School of Medicine. Informed consent was waived for this retrospective study. We reviewed data for 5133 patients with gastric adenocarcinoma who underwent surgical resection or immunotherapy. The inclusion criteria for the surgical cohorts were: histologically confirmed diagnosis of GC; at least 15 lymph nodes harvested; preoperative contrast-enhanced abdominal CT available; and complete clinicopathological and follow-up data available. We excluded patients whose primary tumor could not be identified on CT, who received neoadjuvant chemotherapy or had other synchronous malignant neoplasms. For the immunotherapy cohort, the inclusion criteria were: pretreatment contrast-enhanced abdominal CT available, and clinicopathological and follow-up data available.

A total of 2799 patients in seven independent cohorts were enrolled in this study (Supplementary Fig. 1). In the Nanfang Hospital cohort, we divided patients into training and validation cohorts by the time of surgery. The training cohort and two internal validation cohorts included 348, 202, and 636 patients who were consecutively treated at Nanfang Hospital of Southern Medical University (Guangzhou, China) from January 1, 2005 to December 31, 2008, from January 1, 2009 to June 30, 2012, and from July 1, 2012 to December 31, 2016 respectively. Of note, the training cohort contains patients with complete data available that are necessary for model development.

The two external validation cohorts included 125 and 1062 patients consecutively treated at Sun Yat-sen University Cancer Center (SYSUCC) between June 1, 2007 and June 30, 2013. Another international external validation cohort included 123 patients treated at Stanford University Medical Center between August 1, 2000 and May 31, 2013. Additionally, we enrolled advanced GC patients treated with anti-PD1 immunotherapy at two institutions between January 1, 2019 and July 31, 2021.

Clinicopathologic data including age, gender, tumor and lymph node status, tumor differentiation, Lauren histology type, carcinoembryonic antigen (CEA), and cancer antigen 19-9 (CA19-9) were collected. D2 lymph node dissection was performed in most patients (>90%) in accordance with the Japanese guidelines[41]. All patients were restaged according to the eighth edition of the American Joint Committee on Cancer (AJCC) staging criteria. In the training cohort, internal validation cohorts 1 and 2 from Nanfang Hospital, there were 173 (49.70%), 92 (45.5%), and 373 (58.6%) patients who received 5-fluorouracil–based chemotherapy, respectively. In the external validation cohort from SYSUCC, 559 (47.9%) patients received 5-fluorouracil–based chemotherapy.

The immunotherapy cohort consists of 303 patients with advanced GC treated at Nanfang Hospital and Guangdong Provincial Hospital of Chinese Medicine. Anti-PD-1 drugs include: Nivolumab, Pembrolizumab, and Toripalimab. Clinical data, including patient demographics, treatment information, laboratory & pathologic examinations, and computed tomography (CT) scans were acquired. Microsatellite instability (MSI) status was assessed by either IHC or DNA sequencing.

## Definition of TME classes

We defined four TME classes using two previously validated immune and stromal biomarkers in gastric cancer, i.e., the ImmunoScore of Gastric Cancer ($IS_{GC}$)[42] and protein expression of periostin (POSTN). The $IS_{GC}$ score consists of several important immune cell types and is calculated as: $IS_{GC}$ = (0.149*CD3$_{invasive\ margin}$) + (0.021*CD3$_{center\ of\ tumor}$) + (0.044*CD8$_{invasive\ margin}$) + (0.096*CD45RO$_{center\ of\ tumor}$) − (0.173*CD66b$_{invasive\ margin}$). On the other hand, periostin is an extracellular matrix protein secreted predominantly by stromal cells, which regulates cancer cell migration, invasion, metastatic dissemination, and

chemoresistance[43–45]. These biomarkers represent two major axes of the TME and therefore were used to define TME classes in this study (Fig. 1). Specifically, both the $IS_{GC}$ score and POSTN were dichotomized into low vs. high expression according to their medium values in the training cohort. By doing so, we further divided patients into 4 classes as follows: TME class 1 (high $IS_{GC}$/low POSTN), TME class 2 (high $IS_{GC}$/high POSTN), TME class 3 (low $IS_{GC}$/low POSTN), TME class 4 (low $IS_{GC}$/high POSTN). The assessment of these two biomarkers were described in the following section.

## Immunohistochemistry assessment of TME biomarkers

Formalin-fixed paraffin-embedded (FFPE) tumor tissue samples were processed for immunohistochemistry (IHC) staining. The samples were incubated with antibodies against human CD3, CD8, CD45RO, CD66b, and POSTN (Supplementary Table 3). Prior to staining, sections were blocked with endogenous peroxidase (prepared in 1% $H_2O_2$/methanol solution) for 10 min and then microwaved for 30 min in 10 mM citrate buffer, pH 6.0. The sections were blocked using 10% normal rabbit serum for 30 min. All slides were stained with the same concentrations of primary antibody for each antibody and incubated with monoclonal primary antibody overnight at 4 °C, followed by incubation with an amplification system with a labeled polymer/HRP (EnVision™, DakoCytomation, Denmark) at 37 °C for 30 min. The sections were developed with 0.05% 3, 3´-diaminobenzidine tetrahydrochloride (DAB) and counterstained with modified Harris hematoxylin. Every staining run contained a slide treated with phosphate buffer saline buffer in place of the primary antibody as a negative control.

IHC evaluation was independently performed by two gastroenterology pathologists (T.L. and S.X. with 5 to 10 years of experience) who were blinded to the outcome data. In cases where differences arose between the two primary pathologists, a third pathologist was consulted to reach a consensus. At low power (100x), the tissue sections were screened using an inverted research microscope (model DM IRB; Leica, Germany). Two areas of interest, i.e., center of tumor (CT) and invasive margin (IM), were evaluated at 200x magnification to measure the density of stained immune cells for $IS_{GC}$ calculation. The nucleated stained cells in each area were quantified and expressed as the number of cells per field. For a representative analysis of POSTN, five high-power fields sampled randomly over the entire tumor area in the total specimen were selected for evaluation at 200× magnification. Stain intensity was graded as 0 (negative staining), 1 (weak staining), 2 (moderate staining), and 3 (strong staining); stain extent was graded as 0 (0–4%), 1 (5–24%), 2 (25–49%), 3 (50–74%), and 4 (>75%)[45]. Values of the stain intensity and extent were multiplied and then averaged over the five fields as the final score for each marker.

## CT acquisition and image processing

Patients underwent contrast-enhanced abdominal CT scans prior to treatment. All CT scans were acquired in the cross-sectional (or transaxial) planes, which is the standard abdominal imaging protocol. Following intravenous contrast administration, arterial and portal venous-phase contrast-enhanced CT scans were performed after delays of 28 s and 60 s, respectively. Iodinated contrast material in the amount of 90–100 ml (Ultravist 370, Bayer Schering Pharma, Berlin, Germany) was injected at a rate of 3.0 or 3.5 ml/s with a pump injector (Ulrich CT Plus 150, Ulrich Medical, Ulm, Germany). The type of CT scanners included GE Lightspeed 16, GE Healthcare Milwaukee, WI; 64-section LightSpeed VCT, GE Medical Systems, Milwaukee, WI; USA. The CT acquisition protocols were as follows: 120 kV; 150–190 mAs; 0.5- or 0.4-second rotation time. Contrast-enhanced CT was reconstructed with a field of view, 350 × 350 mm; data matrix, 512 × 512; in-plane spatial resolution 0.607–0.75 mm; axial slice thickness, 1.25–7.5 mm.

Portal venous-phase CT images were retrieved from the picture archiving and communication system (Carestream, Canada). The images used were in their native DICOM format. CT images were resampled to a consistent spatial resolution of $0.75 \times 0.75 \times 2.5$ mm by using trilinear interpolation. We normalized the CT intensity to a window of $[-150, 150]$ HU to highlight the soft-tissue contrast. To focus analysis on the most relevant region (i.e., gastric carcinoma), we delineated the primary tumor as the region of interest. This was performed by two radiologists (C.C. and Q.Y. with 11 and 10 years of clinical experience in abdominal CT interpretation, respectively) using the ITK-SNAP software. Patients whose tumors cannot be identified on CT scans, for example, small tumors <1 cm or highly infiltrative tumors with linitis plastica, were excluded. These patients represent a small percentage of the total population, and their clinical behavior and prognostic patterns are relatively well-defined. Both radiologists were present and reached consensus regarding tumor delineation.

### Development of a deep learning model to predict TME classes and survival

We designed a multi-task deep learning model to simultaneously predict the TME classes and disease-free survival (DFS) from CT images. In multi-task learning, multiple tasks are simultaneously learned by a single model. Intuitively, it can be viewed as a form of inductive transfer, which causes the model to prefer hypotheses that explain more than one task[46]. By sharing representations between related tasks, learning can be made more data efficient and may lead to solutions that generalize better. Indeed, multi-task learning has been shown to reduce overfitting and improve model generalizability in many computer vision applications[47].

Figure 2 shows the flowchart of the proposed network architecture. Here, we used ResNet-18 as the network backbone for feature extraction[48]. We then concatenated the features from all slices and input them into an attention-based module, which allows the network to focus on the most relevant information for prediction. Given the known relations between the two tasks, the network outputs of predicted TME classes are fed as additional input features into the fully connected layer for prediction of DFS.

In detail, we input five CT image slices with a size of $160 \times 160$ centered around the largest tumor section to the ResNet-18 model and obtained five 1-dimensional features of size 256. After that, two fully connected (FC) layers with Leaky Relu as the activation function were employed to refine these features. We then concatenated the features from all slices and input them into an attention-based module. Here, we used two different FC layers to process the above features according to Eq. (1) and obtained an attention map $A$:

$$A = softmax(w_3(\tanh(w_1 f + b_1) * sigmoid(w_2 f + b_2)) + b_3) \quad (1)$$

Where $f$ is the input feature, $w_i f + b_i$ ($i = 1, 2, 3$) denotes the FC layer. Tanh, sigmoid and softmax is the activation function. Note that the dimension of the third FC layer ($w_3 f + b_3$) follows the number of tasks. We multiplied the attention map with $f$ to get the final feature M with size of $N \times 256$, N is the number of tasks.

The proposed deep learning model is simultaneously conducting two tasks: TME classification and DFS prediction. Therefore, we first used an FC layer to process the first-row feature in the final feature $M[0]$ and output the probability of TME classes. Given the known relations between the two tasks, we concatenated the second-row feature in the final feature $M[1]$ with the predicted TME classes and then used two FC layers to output the final DFS.

For training the multi-task learning model, the total loss function $L(\theta)$ is defined as:

$$L(\theta) = L_{TME}(\mathbf{Y}, \widehat{\mathbf{Y}}) + L_{DFS}(\theta) \quad (2)$$

We used cross-entropy as the loss function for predicting the probability of TME class:

$$L_{TME}(\mathbf{Y}, \widehat{Y}) = Y\log(\widehat{Y}) + (1 - Y)\log(1 - \widehat{Y}) \quad (3)$$

where $\widehat{Y}$ is the output of the model and Y is the ground truth of TME class.

For predicting disease-free survival, we used the Cox model to estimate the hazard function. The Cox loss function can be defined as:

$$L_{DFS}(\mathbf{w}, b) = -\log(\text{pl}(\hat{r}_{\mathbf{w},b})) = -\log\left(\prod_{i=1}^{k} \frac{e^{\hat{r}_{\mathbf{w},b}^{i}}}{\sum_{j \in \mathfrak{R}(T_i)} e^{\hat{r}_{\mathbf{w},b}^{j}}}\right) \quad (4)$$

where $\mathfrak{R}(T_i)$ represents the set of patients who are still in the observational study at the time $T_i$. $w$ and $b$ represent the weights and bias parameters of the proposed network, and $\hat{r}_{w,b}$ is the output of the network. $\text{pl}(\cdot)$ represents the partial likelihood of the risk of death observed in the patients by multiplying the conditional probability of the individual patient's death at each time $T_1$, $T_2$, $\cdots$, $T_k$. By substituting network output by the hazard function $\hat{h}_\theta(x)$, the loss function can be simplified as:

$$L_{DFS}(\theta) := -\sum_{i:E_i=1} \left(\hat{h}_\theta(x_i) - \log \sum_{j \in \mathfrak{R}(T_i)} e^{\hat{h}_\theta(x_j)}\right) \quad (5)$$

The proposed framework was implemented on the open-source TensorFlow and trained using the NVIDIA Tesla V100 GPU workstation. We used Adam optimizer to train the proposed framework with a learning rate of $1e - 5$. The batch size for the mini-batch random gradient descent method was set at 16. To mitigate risk of overfitting, we use data augmentation to increase the number and diversity of training samples, which has been shown to improve model generalizability[49]. In brief, we applied data augmentation on the fly during generating training batches by the imgaug toolbox. The augmentation included image reflection along with the patient's anterior/posterior or left/right directions, random affine translation, gaussian blur, image sharpening, and image enhancement with the Laplacian operator.

### Evaluation of the model accuracy for TME classification

We evaluated the accuracy of the CT imaging-based deep learning model to predict the IHC-defined TME classes. Beside the training cohort, the model was tested in two independent cohorts (internal validation cohort 1 and external validation cohort 1) for which IHC data was available. Metrics including the area under the receiver operating characteristic curve (AUC), per-class and overall accuracy were computed. Additionally, we also evaluated the sensitivity, specificity, positive and negative predictive values. The confusion matrix was used to quantify the pairwise classification accuracy among different TME classes.

### Evaluation of the model accuracy for prognosis prediction

We assessed the prognostic accuracy of the deep learning model for predicting the actual events of DFS and OS in terms of discrimination and calibration. DFS was defined as the time from surgery to disease progression or death. OS was defined as the time to death from any cause. AUC was used to evaluate the accuracy for prediction of 5-year DFS and OS. Calibration curves were generated to compare the predicted survival probabilities with the actual probabilities for the event of interest. In addition, we assessed whether the survival model could distinguish patients with distinct prognoses using Kaplan-Meier analysis. The medium value of predicted survival scores in the training cohort was used as the cutoff value, and the same threshold was

applied to the validation cohorts. This analysis was performed for all 2430 patients in 6 independent cohorts (excluding those in the immunotherapy cohort).

An integrated model that combines the imaging signature and clinicopathologic factors was constructed by using the multivariable Cox regression analysis in the training cohort for prediction of DFS and OS. In comparison, a clinicopathologic model was built by including only prognostic clinicopathologic factors, including T stage, N stage, M stage, CEA, tumor location and differentiation. To quantify the relative improvement in prediction accuracy, the net reclassification improvement (NRI) was calculated. The overall performance of these models was assessed by using the prediction error curves and integrated Brier score (IBS).

### Evaluation of the model's association with benefit from chemotherapy

We investigated the deep learning model for its ability to predict the survival benefit of adjuvant chemotherapy in patients with stage II and III GC in a post-hoc exploratory analysis. To minimize selection bias and confounding effects, we used a matching strategy to balance patients in each defined TME class[50]. Specifically, propensity score matching (PSM) was performed for patients who received vs. did not receive adjuvant chemotherapy using 1:1 nearest matching. Propensity scores were calculated using the following clinicopathologic variables: age, sex, differentiation, CEA, CA19-9, location, depth of invasion (T stage), lymph node metastasis (N stage), tumor size, and Lauren type.

### Evaluation of the model's association with response to anti-PD1 immunotherapy

We analyzed an independent cohort of 303 advanced GC patients treated with anti-PD-1 immune checkpoint blockade, and assessed clinical response and outcomes in relation to the deep learning model predicted TME classes. Response to immunotherapy was evaluated according to the irRECIST criteria[51] and defined as complete response (CR), partial response (PR), stable disease (SD), or progressed disease (PD). Objective response was defined for patients who achieved either CR or PR. Progression-free survival (PFS) was calculated from the start of treatment until disease progression, death, or last follow up. The combined positive score (CPS) was defined as the total number of PD-L1 positive cells (tumor, lymphocytes, and macrophages) divided by the total number of viable tumor cells multiplied by 100. CPS was categorized as high ($CPS \geq 10$), intermediate ($10 > CPS \geq 1$), and low ($CPS < 1$). An integrated model that combines CPS and predicted TME was built using the Classification and Regression Tree algorithm.

### Statistical analysis

We compared two groups using the t-test for continuous variables and the chi-square test or Fisher exact test for categorical variables, as appropriate. Comparisons in AUC were performed using the DeLong's method. Survival curves were generated according to the Kaplan-Meier method and compared using the log-rank test. Univariate and multivariate analyses for survival were performed with the Cox proportional hazard model. All clinicopathological variables were used for the multivariate analysis. The relative importance of each parameter to survival risk was assessed using the $\chi^2$ from Harrell's rms R package. Interaction between the TME classes/survival score and chemotherapy was also assessed by the Cox model. The "survival ROC" package was used to perform the time-dependent ROC analysis. Nomograms and calibration plots were generated using the "rms" package in R. Net reclassification improvement was computed using the "survIDINRI" package in R. The prediction error curves were obtained using the "pec" package in R ("Boot-632plus" split method with 1000 iterations). $P$ value less than 0.05 was defined as statistically significant in two-

tailed analyses. Statistical analyses were performed using R version 4.1.0 and SPSS version 21.0.

### Reporting summary

Further information on research design is available in the Nature Portfolio Reporting Summary linked to this article.

## Data availability

Source data are provided with this paper. The source data underlying Figs. 3–8, Supplementary Figs. 3–23, and Supplementary Table 1, 2, 4–11, 13, 22 and 23 are provided as a Source Data file. The remaining data are available within the Article, Supplementary Information, or Source Data file. The de-identified individual patient data including CT images, tumor segmentations, IHC evaluation, clinicopathologic and follow-up data are available. A data transfer agreement is required that includes a brief research plan submitted by the user and data usage is restricted to non-commercial academic research purposes. Request for data access can be submitted to R.L. and will receive a response typically within 10 days. Data will be shared through cloud storage and available for 1 year once access has been granted. Source data are provided with this paper.

## Code availability

Source code for the deep learning model is available at: https://github.com/zzc623/ClassGastric. Source code has also been placed on the Zenodo platform [https://zenodo.org/record/8176377][52].

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

## Acknowledgements

This study was supported in part by the U.S. National Institute of Health under a research grant 1R01CA269559 (R.L.).

## Author contributions

R.L., G.L., and Y.J. conceived and designed the study; Y.J., C.C., W.W., T.L., W.H., Z.H., M.U.A., Y.R. and W.X. acquired the data; Y.J., J.X., S.X., J.W., W.H. and Z.Z. did the statistical analyses; Y.J., Z.Z. and S.S. developed, trained, and applied the artificial neural network. R.L., L.X., G.P., Y.X. and G.L. implemented quality control of data and the algorithms; Y.J., C.C., Q.Y., W.X. and W.W. verified the underlying raw data; All authors had access to the data presented in the manuscript. All authors analyzed and interpreted the data; Y.J., Z.Z., W.W. and W.H. prepared the first draft of the manuscript; R.L. revised the manuscript; All authors contributed to manuscript preparation.

## Competing interests

The authors declare no competing interests.

## Additional information

**Supplementary information** The online version contains
supplementary material available at

Guoxin Li or Ruijiang Li.

**Peer review information** *Nature Communications* thanks Enrico Capo-
bianco, Issam El Naqa and the other, anonymous, reviewer(s) for their
contribution to the peer review of this work. A peer review file is avail-
able.

