## [Peer Review file · Nature Communications]

REVIEWER COMMENTS

Reviewer #1, expertise in ML and radiomics (Remarks to the Author):

The manuscript presented a multi-task deep learning algorithm to predict TME status and immunotherapy response in a population of gastric cancer patients from CT images. The study seems an extension to their previous work (Jiang et al, lancet digital health, 2021).

- can you detail what is the objective function used for multi-task learning?
- The AUCs seem similar to (Jiang et al, lancet digital health), is this right?
- Why AUC is used is measuring survival performance? Is the end point classification only at 5 years?
- How the biology of TME was used? Isn't the information derived from CT only?
- The manuscript seems to mention the shortcoming of saliency maps, nevertheless applies them? How the multi-task and the sequential design varied here?
- There is a reference to the code, would the data be made available too?

Reviewer #2, expertise in gastric cancer TME and immunotherapy (Remarks to the Author):

This is an article by Jiang and colleagues from Southern Medical University, Stanford University and Sun Yat-sen University Cancer Center, performing an analysis using deep learning model to predict TME and prognosis as well as treatment outcomes from radiological images. While the study is interesting, several issues need to be considered.

A lot more detail needs to be provided on which parts of the CT images were used and correlated to histology images? Which cut of the CT was used and why? Or were multiple cuts used? Were they sagittal or cross-sectional imaging? Was this standardized? Very often it is almost impossible to visualize a stomach tumor on a CT scan, especially early stages. Then in this case, what did the investigators use? What were the steps taken to ensure that the same level of the CT cuts were correlated with histological sample? It is non-trivial to correlate a biopsy taken from an endoscopy or surgical resection and sample and match that to a CT scan. It is not clear to this Reviewer how the TME classification was associated with the deep learning model? Was the classification done manually or by machine learning?

How many pathologists were involved in the TME classification. Was this done independently? How was conflicts resolved in TME assessments.

Can the authors suggest what component of the imaging is guiding the prediction? Could it just be size, which correlates with T stage? Who is demarcating the tumor on the scan images? Is this done by ML or manually by a radiologist?

Given that the stage is a very good determinant of prognosis, similar to machine learning, how do we know that the machine learning model isn't merely just measuring the tumor size and using this to determine prognosis? What other features other than size is the model using? How do we know that the model isn't merely just using the clinicopathological characteristics that are already readily available in determining prognosis?

For the prediction of benefit for adjuvant chemotherapy, a multivariate analysis should be performed to determine if the TME or DLS are independent predictors of benefit from chemotherapy. If so, this point can be further expounded as it has significant clinical implications and value. Simply put, by applying a ML predictor on a pre-surgery CT scan, if we can better predict for which patients will benefit from adjuvant chemotherapy, this will be very valuable to clinicians. Again, the question will be, how does one upload the scans, what sorts of scans are acceptable, what are the parameters that are required, and does the tumor area need pre-demarcation by a radiologist or can the ML model identify the tumor directly. These need to be clarified.

For the immunotherapy treatment -> were these patients treated with single agent or combination with chemo? Or mixed? If it is mixed, then the analysis needs to be divided to study single agent and combination separately. It is unclear on how heterogeneous this group of patients who were treated with immunotherapy are.

Reviewer #3, expertise in ML and radiomics (Remarks to the Author):

The results presented in this paper are valuable, in principle.

Therefore, the work is potentially significant.

The methodology is sound,

The authors claim superior results compared to the literature.

In my opinion, the work support partially the conclusions and claims, and additional evidence is needed.

The data analysis is fine, but interpretation and conclusions are in part questionable. I therefore require revision.

There is a general lack of details (data more than code) for the work to be reproduced?

Specific comments.

citing the authors:

It is important to distinguish two related but different concepts in machine learning: interpretability and explainability, although they are often used interchangeably. In this work, we view 'interpretability' as models being constrained by biological knowledge or mechanism. On the other hand, 'explainability' relates to how a model works, e.g., by finding specific features

that are simple and intuitive to a human.

While both are important aspects of a machine

learning model, we are more concerned about interpretability, i.e., why a model works for highstake clinical applications.

Here, we show that incorporation of biological domain knowledge

about TME as integral components of deep learning not only improved generalizability compared with the traditional approach but also enhanced interpretability of the model.

My comment:

The choice of being more concerned with interpretability than explainability appears a bit tranchant.

These two aspects should go together when there are reasons for which explainability may be required as part of the proposed approach.

If the authors have a strategy for dealing with explainability, and thus determining criteria to select significant features, they should describe it. Otherwise, the parallel is weak.

If the authors do not deal with explainability, and indeed they do not deal with it, then better not to discuss and focus on their concern.

They write:

Figure 2B shows the CT images and corresponding feature maps along with the predicted TME classes and survival scores for four representative cases. As can be seen from the feature maps,

the deep learning model captures important information related to intratumor heterogeneity as well as imaging characteristics of the invasive margin (Figure 2B).

My first comment:

only by explaining the features used to build the maps one can infer whether heterogeneity is captured or not.

I am not seeing this element in the paper.

They write:

Several groups including ours have developed radiological signatures of immune biomarkers that were subsequently correlated with treatment response and outcomes 28-32.

Our study is distinct from prior work in several aspects.

First, previous studies have mostly focused on tumor-infiltrating lymphocytes^{28,29}, which only provides a simplified and partial view of the TME.

My comment:

This again is a style that tends to focus on something not covered in this paper, de facto diminishing the current landscape of radiomic applications involving TME across multiple characteristics.

It should be avoided, the authors should stick with what they demonstrate and not with conjectures or what is not part of this work.

They write:

Although we focused on specific aspects of the tumor microenvironment in this work, the proposed framework is general in that any relevant information about cancer biology can be incorporated to design knowledge-guided deep learning

models for prediction of treatment response and outcomes. As an alternative approach to deep learning, traditional radiomics rely on hand-crafted image features and may offer some degree of interpretability³⁶⁻³⁹. In future work, it may be beneficial to combine the two complementary approaches to further enhance model performance and interpretability.⁴⁰

My comment:

The fact that traditional radiomics relies on hand crafted image features is not completely true. This is true for a minority of studies. The majority relies on ML and DL or statistical approaches.

It is therefore an oversimplification what the authors present in this point, and I disagree with it because not true.

Finally:

Data Availability Statement

Deidentified patient-level clinical data will be provided upon reasonable request.

The CT image data are not publicly available because they contain sensitive information that could compromise

patient privacy.

My comment:

I am disappointed about this statement.

The data should be available to reproduce these results.

The CT images, the segmentations and the annotations to identify the ROIs should be set available, like many other images appearing in major retrospective studies and repositories like TCIA.

The explanation offered is not acceptable as the images can be easily subjected to multiple software to remove patient information.

The segmentations should already appear without this component.

Once again, without a plan to put these data and images available, the results here presented lack of reproducibility and diminish the value the authors assign to their approach consistently claimed as superior in comparative terms.

Unless the authors validate their results also over publicly available data-image sets.

Response to Reviewer Comments

Jiang et al. "Biology-guided Deep Learning Predicts Prognosis and Cancer Immunotherapy Response"

We sincerely appreciate the thorough review and constructive comments and suggestions by the reviewers. We have responded to the reviewer comments in detail and revised the manuscript accordingly. Our point-to-point response is shown below in blue text.

REVIEWER COMMENTS

Reviewer #1, expertise in ML and radiomics (Remarks to the Author):

The manuscript presented a multi-task deep learning algorithm to predict TME status and immunotherapy response in a population of gastric cancer patients from CT images. The study seems an extension to their previous work (Jiang et al, lancet digital health, 2021).

Thank you for this comment. We highlight the conceptual and translational advances of the current work over previous studies including our previous publication (Jiang et al, Lancet Digital Health, 2021).

- We incorporate biological knowledge into the model, which showed superior performance to traditional data-driven approach and our previous work (Jiang et al. 2021) for outcome prediction.
- The model not only predicts prognosis of over 2500 patients from Asia, but also is generalizable to a US-based population with multiple racial and ethnic groups. Our previous study (Jiang et al. 2021) did not include any patients outside Asia.
- The model predicts clinical response to immune checkpoint inhibitors (ICI). Importantly, our model identified a subset of patients with mismatch repair-deficient tumors who did not respond well to immunotherapy. This is a novel finding with translational impact because these patients are clinically approved to receive ICI.
- The immunotherapy dataset of 303 patients by itself is one of the largest cohorts reported in the literature for gastric cancer. By contrast, our previous study (Jiang et al. 2021) did not include any patients treated with immunotherapy.

- can you detail what is the objective function used for multi-task learning?

For training the multi-task learning model, the objective function is defined as the summation of TME classification error and prognosis prediction error. We used cross-entropy as the loss function for TME classification and Cox loss function for prognosis prediction. This has been described in detail in supplementary methods, under section 'Model training and implementation', and equations (2)-(5).

- The AUCs seem similar to (Jiang et al, lancet digital health), is this right?

The two studies have different goals and tasks. The purpose of our previous study is for stromal TME classification. On the other hand, the goal of our current work is combined immune/stromal TME classification and prognosis prediction. Therefore, we cannot make a direct comparison between the AUCs for TME classification.

However, we did show a superior performance for prognosis prediction using the proposed model over our previous work based on stroma status (Figure S15).

- Why AUC is used is measuring survival performance? Is the end point classification only at 5 years?

The survival prediction model is trained for estimating the hazard function in a Cox regression. The model is capable of making predictions at multiple time points. Here, we focused on the 5-year survival as the primary endpoint, which is widely used for evaluating prognostic models. In our revision, we have also reported the C-index for the deep learning model, clinicopathologic variables, and integrated models. The results show similar patterns when comparing different variables and models (Figure S11).

Figure S11. C-index of prediction for disease-free survival (DFS) using clinicopathologic variable and the deep learning model. C-index: concordance index.

- How the biology of TME was used? Isn't the information derived from CT only?

The ground truth for TME status was defined based on IHC evaluation of several immune and stromal markers as detailed in our methods.

The TME classification is then used as an integral component of the deep learning model (one branch for the output) to guide the outcome prediction from CT images, as shown in Figure 1.

- The manuscript seems to mention the shortcoming of saliency maps, nevertheless applies them? How the multi-task and the sequential design varied here?

The sequential approach, which was used in our previous work (Jiang et al, Lancet Digital Health, 2021), is designed to predict specific biological features, and then assess their relation to clinical outcome separately in a post hoc analysis. However, this approach is biased by prior knowledge and is unable to identify other unknown factors that may influence treatment outcome.

Here, we incorporate the tumor microenvironment into model design and training process via a multi-task learning strategy for simultaneous prediction of TME and outcome. This allows us to identify novel features beyond TME that are associated with outcome. Indeed, we showed the multi-task learning model outperformed CT image-based prognosis deep learning model and single-task TME model for prognosis prediction in Figure S15B, S16.

The saliency maps are often used as post hoc explanation after the model has already been trained/locked-down. In Figure 2, we showed the feature maps directly taken from the network, which are different from saliency maps.

- There is a reference to the code, would the data be made available too?

Yes, we will make the data available for patients in the training, internal and external validation cohorts. These include the CT images (ROI), tumor segmentations, IHC evaluation of TME status, clinicopathologic data and follow-up information.

Reviewer #2, expertise in gastric cancer TME and immunotherapy (Remarks to the Author):

This is an article by Jiang and colleagues from Southern Medical University, Stanford University and Sun Yat-sen University Cancer Center, performing an analysis using deep learning model to predict TME and prognosis as well as treatment outcomes from radiological images. While the study is interesting, several issues need to be considered.

A lot more detail needs to be provided on which parts of the CT images were used and correlated to histology images? Which cut of the CT was used and why? Or were multiple cuts used? Were they sagittal or cross-sectional imaging? Was this standardized? Very often it is almost impossible to visualize a stomach tumor on a CT scan, especially early stages. Then in this case, what did the investigators use?

All CT scans were acquired in the cross-sectional (or transaxial) planes, which is the standard abdominal imaging protocol. The images used were in their native DICOM format. In this work, five consecutive image slices centered around the largest tumor section were input to the deep learning model. Patients whose tumors cannot be identified on CT scans, for example, small tumors <1 cm or highly infiltrative tumors with linitis plastica, were excluded. These patients represent a small percentage of the total population, and their clinical behavior and prognostic patterns are relatively well-defined. We have clarified these points in the Methods section.

What were the steps taken to ensure that the same level of the CT cuts were correlated with histological sample? It is non-trivial to correlate a biopsy taken from an endoscopy or surgical resection and sample and match that to a CT scan. It is not clear to this Reviewer how the TME classification was associated with the deep learning model? Was the classification done manually or by machine learning?

An important advantage of imaging is that it allows noninvasive characterization of the entire tumor, which overcomes the challenge of sampling bias with tissue-based histopathology approach. Therefore, our strategy in this study is to estimate the TME status at the tumor level based on imaging analysis of the whole tumor. This is achieved by training a deep learning model using a cohort of 348 patients who had matched imaging data and tumor tissue stained with IHC for TME classification available (as shown in Figure 1).

The clinical implication of this work is that, once an imaging-based model of TME is trained, it can be applied to new patients to assess TME based on imaging analysis of the whole tumor, noninvasively and longitudinally. This will overcome limitations of the current approach of biopsy which is prone to sampling bias and often not repeated due to its invasive nature.

We did not attempt to spatially correlate CT images and histology mainly for two reasons. First, it is nearly impossible to accurately co-register the two images at the pixel level due to vast differences in resolution, tissue deformation, and other changes; errors made in registration will adversely affect model performance. Second, even if it is feasible, such an approach would not be practical since it always requires tissue specimen to be matched to CT when applying to new patients. Our model only requires CT images as input for its applications.

How many pathologists were involved in the TME classification. Was this done independently? How was conflicts resolved in TME assessments.

IHC evaluation was independently performed by two gastroenterology pathologists (T.L. and S.X. with 5 to 10 years of experience) who were blinded to the outcome data. In cases where differences arose between the two primary pathologists, a third pathologist was consulted to reach a consensus. We have clarified this point in the Methods section.

Can the authors suggest what component of the imaging is guiding the prediction? Could it just be size, which correlates with T stage? Who is demarcating the tumor on the scan images? Is this done by ML or manually by a radiologist?

We observed that imaging features related to intratumor heterogeneity and invasive margin are important characteristics for prediction as shown in Figure 2. To support these findings, we have provided additional evidence using quantitative radiomics features to measure intratumor heterogeneity in our revision.

Although the deep learning model (survival score) was statistically associated with tumor size, the magnitude of their correlation was low (Pearson correlation coefficients 0.14- 0.17).

The primary tumor was delineated on the CT images by two radiologists (C.C. and Q.Y. with 11 and 10 years of clinical experience in abdominal CT interpretation, respectively) using the ITK-SNAP software.

Given that the stage is a very good determinant of prognosis, similar to machine learning, how do we know that the machine learning model isn't merely just measuring the tumor size and using this to determine prognosis? What other features other than size is the model using? How do we know that the model isn't merely just using the clinicopathological characteristics that are already readily available in determining prognosis?

We showed in multivariable Cox regression analysis that the deep learning model is a strong prognostic factor independent of tumor stage and other clinicopathological characteristics across five validation cohorts. Please refer to Table 2.

For the prediction of benefit for adjuvant chemotherapy, a multivariate analysis should be performed to determine if the TME or DLS are independent predictors of benefit from chemotherapy. If so, this point can be further expounded as it has significant clinical implications and value. Simply put, by applying a ML predictor on a pre-surgery CT scan, if we can better predict for which patients will benefit from adjuvant chemotherapy, this will be very valuable to clinicians.

Thank you for this valuable suggestion. We have performed multivariate logistic regression analysis and confirm that the imaging based TME class is indeed an independent factor for predicting benefit of adjuvant chemotherapy in gastric cancer. The result is shown in the following table.

Table S22. Multivariate analysis for association with chemotherapy response.

TME class		
1	Reference	<0.0001
2	0.257 (0.138-0.480)	<0.0001
3	0.170 (0.093-0.312)	<0.0001
4	0.054 (0.028-0.105)	<0.0001
Depth of invasion		.027
pT1	Reference	
pT2	0.604 (0.341-1.068)	.083
pT3	0.724 (0.416-1.257)	.251
pT4a	0.627 (0.371-1.060)	.082
pT4b	0.393 (0.222-0.693)	.001
Lymph node metastasis		.050
pN0	Reference	
pN1	0.620 (0.158-2.427)	.492
pN2	0.354 (0.103-1.220)	.100
pN3a	0.396 (0.117-1.335)	.135
pN3b	0.239 (0.068-0.841)	.026

Again, the question will be, how does one upload the scans, what sorts of scans are acceptable, what are the parameters that are required, and does the tumor area need pre-demarcation by a radiologist or can the ML model identify the tumor directly. These need to be clarified.

The only requirement for applying the model is a contrast-enhanced CT scan, which is part of standard clinical care. The model has been trained and tested on scans from different manufactures with various imaging protocol parameters and the performance is generalized well. At present, manual delineation of the tumor is required for the best model performance.

This is mainly because with current ML it is extremely challenging to reliably segment gastric cancer due to the inherent nature of these tumors, such as irregular shapes and indistinct boundary present in the CT images. In future, development of more advanced methods for automated or semi- automated tumor segmentation can help facilitate practical implementation of this approach. We have discussed this issue in our revision.

For the immunotherapy treatment -> were these patients treated with single agent or combination with chemo? Or mixed? If it is mixed, then the analysis needs to be divided to study single agent and combination separately. It is unclear on how heterogenous this group of patients who were treated with immunotherapy are.

In our original analysis, the majority of patients (N=235) received anti-PD1 immunotherapy combined with chemotherapy; the remaining 18 patients received single-agent immunotherapy. This cohort reflects the current treatment landscape in advanced gastric cancer where patients typically receive combination therapies.

We agree that it is important to evaluate the therapeutic effect in patients treated with immunotherapy only. Since the initial data collection for this study, we have identified 50 additional patients who were treated with single-agent immunotherapy. Combined with the original 18 patients, this provided a total of 68 patients treated with single-agent immunotherapy, which allowed us to confirm the relation between the DL model and effect of immunotherapy.

Similar to our previous results, we found the DL model was predictive of response in the single-agent immunotherapy cohort (please see figures below). We have updated the main Figure 7 for all 303 patients treated with immunotherapy with or without chemotherapy. We added the new results for 68 patients treated with single-agent immunotherapy and 235 patients treated with combination therapy in Figures S22 and S23 of the supplementary materials. We have reproduced the figures here for your convenience.

E Multivariate logistic regression analysis for objective response.

Figure S22. Performance of the deep learning model in predicting response and outcomes in patients treated with single-agent anti-PD-1 immunotherapy.

(A), Response rates in patients of four TME classes predicted by the deep learning model; (B), Progression-free survival in patients of four predicted TME classes; (C), Receiver operator characteristic (ROC) curves of the predicted TME classes, CPS and composite models combining TME classes and CPS for predicting immunotherapy response (n=68); (D), AUC values of the predicted TME classes, CPS and composite models combining TME classes and CPS for predicting immunotherapy response (n=68); (E), Forest plot for the multivariate logistic regression analysis for objective response; (F), Decision tree combining the predicted TME classes and CPS. AUC: area under the receiver operator characteristic curve. CPS: combined positive score of PDL1 expression. OR: objective response (complete and partial response); SD: stable disease; PD: progressive disease.

E Multivariate logistic regression analysis for objective response.

Subgroup	Number of patients	OR (95% CI)	P value
TME Class			
1 VS. 3/4	45/141	10.991 (4.951 - 24.402)	<0.001
2 VS. 3/4	42/141	6.661 (3.054 - 14.529)	<0.001
CPS			
Moderate VS. Low	116/42	2.062 (0.785 - 5.416)	0.142
High VS. Low	70/42	4.411 (1.607 - 12.110)	0.004

Figure S23. Performance of the deep learning model in predicting response and outcomes in patients treated with combination therapy of anti-PD-1 immunotherapy and chemotherapy.

(A), Response rates in patients of four TME classes predicted by the deep learning model; (B), Progression-free survival in patients of four predicted TME classes; (C), Receiver operator characteristic (ROC) curves of the predicted TME classes, CPS and composite models combining TME classes and CPS for predicting immunotherapy response (n=228); (D), AUC values of the predicted TME classes, CPS and composite models combining TME classes and CPS for predicting immunotherapy response (n=228); (E), Forest plot for the multivariate logistic regression analysis for objective response; (F), Decision tree combining the predicted TME classes and CPS. AUC: area under the receiver operator characteristic curve. CPS: combined positive score of PDL1 expression. OR: objective response (complete and partial response); SD: stable disease; PD: progressive disease.

E Multivariate logistic regression analysis for objective response.

Subgroup	Number of patients	OR (95% CI)	P value
TME Class			
1 VS. 3/4	58/181	11.008 (5.434 - 22.299)	<0.001
2 VS. 3/4	57/181	6.371 (3.234 - 12.550)	<0.001
CPS			
Moderate VS. Low	149/55	1.664 (0.723-3.835)	0.231
High VS. Low	92/55	4.568 (1.908-10.936)	0.001

Figure 7. Performance of the deep learning model in predicting response and outcomes in patients treated with anti-PD-1 immunotherapy.

(A), Response rates in patients of four TME classes predicted by the deep learning model; (B), Progression-free survival in patients of four predicted TME classes; (C), Receiver operator characteristic (ROC) curves of the predicted TME classes, CPS and composite models combining TME classes and CPS for predicting immunotherapy response (n=296); (D), AUC values of the predicted TME classes, CPS and composite models combining TME classes and CPS for predicting immunotherapy response (n=296); (E), Forest plot for the multivariate logistic regression analysis for objective response; (F), Decision tree combining the predicted TME classes and CPS. AUC: area under the receiver operator characteristic curve. CPS: combined positive score of PDL1 expression. OR: objective response (complete and partial response); SD: stable disease; PD: progressive disease.

Reviewer #3, expertise in ML and radiomics (Remarks to the Author):

The results presented in this paper are valuable, in principle.
Therefore, the work is potentially significant.
The methodology is sound,

The authors claim superior results compared to the literature.
In my opinion, the work support partially the conclusions and claims, and additional evidence is needed.

The data analysis is fine, but interpretation and conclusions are in part questionable. I therefore require revision.

There is a general lack of details (data more than code) for the work to be reproduced?

Specific comments.

citing the authors:

It is important to distinguish two related but different concepts in machine learning: interpretability and explainability, although they are often used interchangeably. In this work, we view ‘interpretability’ as models being constrained by biological knowledge or mechanism. On the other hand, ‘explainability’ relates to how a model works, e.g., by finding specific features that are simple and intuitive to a human. While both are important aspects of a machine learning model, we are more concerned about interpretability, i.e., why a model works for high-stake clinical applications.

My comment:

The choice of being more concerned with interpretability than explainability appears a bit tranchant. These two aspects should go together when there are reasons for which explainability may be required as part of the proposed approach.

If the authors have a strategy for dealing with explainability, and thus determining criteria to select significant features, they should describe it. Otherwise, the parallel is weak.

If the authors do not deal with explainability, and indeed they do not deal with it, then better not to discuss and focus on their concern.

We have removed the above statement about explainability from the Discussion.

They write:

Figure 2B shows the CT images and corresponding feature maps along with the predicted TME classes and survival scores for four representative cases. As can be seen from the feature maps, the deep learning model captures important information related to intratumor heterogeneity as well as imaging characteristics of the invasive margin (Figure 2B).

My first comment:

only by explaining the features used to build the maps one can infer whether heterogeneity is captured or not.

I am not seeing this element in the paper.

We clarify that the feature maps were taken directly from the deep neural networks after model training. They were not prespecified but rather automatically learned during the training process. To understand what type of information the feature maps represent, we computed radiomics features based on the feature maps using the PyRadiomics package, including First-order statistics, Gray Level Cooccurrence Matrix (GLCM) features, Gray Level Run Length Matrix (GLRLM) features, Gray Level Size Zone Matrix (GLSZM) features, Neighboring Gray Tone Difference Matrix (NGTDM), and Gray Level Dependence Matrix (GLDM) features. We observed significant differences in the texture feature values among the four TME classes. Features that measure heterogeneity generally show increasing patterns from TME class 1 through 4, while those measuring homogeneity (GLCM_InverseVariance) show decreasing patterns (Figure S2). This result indicates that the deep learning model may capture important information related to intratumor heterogeneity.

Figure S2. Texture feature values of feature maps in different TME classes.

The radiomics features are calculated based on the feature maps (15th channel) for each of the predicted TME classes. Features that measure heterogeneity generally show increasing patterns from TME class 1 through 4, while those measuring homogeneity (GLCM_InverseVariance) show decreasing patterns.

They write:

Several groups including ours have developed radiological signatures of immune biomarkers that were subsequently correlated with treatment response and outcomes. Our study is distinct from prior work in several aspects. First, previous studies have mostly focused on tumor-infiltrating lymphocytes^{28,29}, which only provides a simplified and partial view of the TME.

My comment:

This again is a style that tends to focus on something not covered in this paper, de facto diminishing the current landscape of radiomic applications involving TME across multiple characteristics.

It should be avoided, the authors should stick with what they demonstrate and not with conjectures or what is not part of this work.

We have removed the above statement about prior studies and instead focus on the significance and novelty of the current work in the Discussion.

They write:

Although we focused on specific aspects of the tumor microenvironment in this work, the proposed framework is general in that any relevant information about cancer biology can be incorporated to design knowledge-guided deep learning models for prediction of treatment response and outcomes. As an alternative approach to deep learning, traditional radiomics rely on hand-crafted image features and may offer some degree of interpretability³⁶⁻³⁹. In future work, it may be beneficial to combine the two complementary approaches to further enhance model performance and interpretability.⁴⁰

My comment:

The fact that traditional radiomics relies on hand crafted image features is not completely true. This is true for a minority of studies. The majority relies on ML and DL or statistical approaches.

It is therefore an oversimplification what the authors present in this point, and I disagree with it because not true.

We realize that the word 'hand crafted' may have narrower meaning and could cause confusion. Therefore, we have replaced it with 'user-defined computational image features'. This term is used to make a distinction from data-driven deep learning models that learn feature representations automatically.

Data Availability Statement

Deidentified patient-level clinical data will be provided upon reasonable request.

The CT image data are not publicly available because they contain sensitive information that could compromise patient privacy.

My comment:

I am disappointed about this statement. The data should be available to reproduce these results. The CT images, the segmentations and the annotations to identify the ROIs should be set available, like many other images appearing in major retrospective studies and repositories like TCIA. The explanation offered

is not acceptable as the images can be easily subjected to multiple software to remove patient information. The segmentations should already appear without this component.

Once again, without a plan to put these data and images available, the results here presented lack of reproducibility and diminish the value the authors assign to their approach consistently claimed as superior in comparative terms.

We appreciate your comment about the data availability. We agree with you that it is important to share research data and make sure that the findings are reproducible. Therefore, we have made the data available for patients in the training, internal and external validation cohorts. These include the CT images (ROI), tumor segmentations, IHC evaluation of TME status, clinicopathologic data and follow-up information. This is sufficient to reproduce the deep learning model training and its validation.

REVIEWERS' COMMENTS

Reviewer #1 (Remarks to the Author):

Thank you for addressing my comments adequately. I still think a more in-depth analysis of findings in contrast with your previous work would be helpful.

Reviewer #2 (Remarks to the Author):

The authors have addressed all my concerns satisfactorily

Reviewer #3 (Remarks to the Author):

Ok with the revision.

Response to Reviewer Comments

REVIEWER COMMENTS

Reviewer #1 (Remarks to the Author):

Thank you for addressing my comments adequately. I still think a more in-depth analysis of findings in contrast with your previous work would be helpful.

Author response: We have compared the proposed multi-task learning approach with the traditional single-task learning approaches by using the immune or stromal TME only as the biological endpoint (approach adopted in our previous work Jiang et al. Lancet Digital Health 2021) for the prediction of disease-free survival. The proposed model significantly improved the AUC for survival prediction over other models by 6-18% in both internal and external validation cohorts. These results are shown in Supplementary Figure 16.

Reviewer #2 (Remarks to the Author):

The authors have addressed all my concerns satisfactorily

Reviewer #3 (Remarks to the Author):

Ok with the revision.